# Is vaccination a viable method to control Johne's disease caused by *Mycobacterium avium* subsp. *paratuberculosis*? Data from 12 million ovine vaccinations and 7.6 million carcass examinations in New South Wales, Australia from 1999–2009

**Ian J. Links**[1]☯*, **Laurence J. Denholm**[2]☯, **Marilyn Evers**[3]☯, **Lloyd J. Kingham**[4], **Robert J. Greenstein**[5]

**1** Graham Centre for Agricultural Innovation (An alliance of Charles Sturt University and NSW Department of Primary Industries), Wagga Wagga, New South Wales, Australia, **2** Strategy and Delivery Group, NSW Department of Premier and Cabinet, Orange, New South Wales, Australia, **3** Formerly NSW Department of Primary Industries, Orange, New South Wales, Australia, **4** NSW Department of Primary Industries, Orange, New South Wales, Australia, **5** James J. Peters VAMC, Bronx, New York, United States of America

☯ These authors contributed equally to this work.

* ijlinks48@gmail.com

## Abstract

### Background

*Mycobacterium avium* subsp. *paratuberculosis* (MAP) causes Johne's disease (or paratuberculosis), a chronic wasting disease of ruminants and other animals resulting from granulomatous enteritis. There are increasing concerns that MAP is zoonotic. The prevalence of Johne's disease is increasing worldwide. In an attempt to control an epidemic of ovine Johne's disease (OJD) in New South Wales (NSW), a government/industry sponsored voluntary vaccination/on-farm management program commenced in 2000. We report herein an observational study of changes in disease prevalence as vaccination progressed, based on abattoir surveillance data for OJD from 1999 to 2009. We also discuss the epidemiological, policy, regulatory, research, economic and sociological elements that contributed to the development of a mature control program, whose aim was to halt the epidemic spread of OJD in a naïve sheep population.

### Methods

NSW was divided into areas of "High" (HPA), "Medium" (MPA) and "Low" (LPA) OJD prevalence. A killed whole cell vaccine (Gudair®) was administered to sheep from 2000 to 2009. Trained examiners evaluated the viscera of adult sheep carcasses at slaughter for gross evidence of OJD. MAP infection was confirmed by histopathology.

### Principal findings

From 2000–2009, 12 million vaccine doses were administered in NSW (91%; 10.9 million in the HPA). Many of the vaccinated flocks were suffering > 5% annual mortality in adult

**Data Availability Statement:** The data has now been deposited in FIGSHARE DOI: 10.6084/m9. figshare.14604198.

**Funding:** Funding was provided under a succession of cost-sharing agreements by multiple agencies including: - New South Wales (NSW) Department of Primary Industries (DPI) representing the NSW State Government and the NSW Sheep Industry (NSW SI) https://www.dpi. nsw.gov.au/animals-and-livestock/sheep/health/ other/ojd https://www.dpi.nsw.gov.au/search? query=ojd Animal Health Australia (AHA) representing the Sheepmeat Council of Australia (SMC) and WoolProducers Australia (WPA). https://www.animalhealthaustralia.com.au/what- we-do/endemic-disease/ovine-johnes-disease-in- australia/ https://sheepproducers.com.au/media- release/best-practice-tools-and-strategies-central- to-on-farm-ojd-control/ The programs were managed by NSW DPI and nationally by AHA. The initial A$40 million National Control and Evaluation Program (NOJDP) ran from 1999 to 2004, the National Approach to the Management of OJD (NAOJD) from 2004 to 2006 and the National OJD Management Plan from 2007 to 2012. Dr Lorna Citer, Manager Endemic Diseases with AHA played a key role as National Coordinator of these programs (including the abattoir monitoring and vaccination programs) from 1999-2009 as outlined in the national management plans. CSL Limited and Pfizer Animal Health provided material support in the form of vaccine supplies; sales data: and strong promotion of safe vaccination procedures. The funders had no role in study design, data collection and analysis, decision to publish, or preparation of the manuscript. AHA and NSW DPI as the program managers provided unrestricted access to the data and did not play any role in the analysis, decision to publish or preparation of the manuscript.

**Competing interests:** The authors have read the journal's policy, and the authors of the manuscript have the following competing interests to declare: CSL Limited and Pfizer Animal Health provided material support in the form of vaccine supplies; sales data: and strong promotion of safe vaccination procedures. The following three relevant patents have been issued to RG: US Patent # 7,846,420: Issue Date Dec 7, 2010. Mycobacterium Avium Subspecies Paratuberculosis Vaccines and Methods for Using the Same. US Continuation-in-part Application entitled "Combination Vaccines Against Mycobacterium Species and Methods of Using Same." Serial #12/956,064 Filing Date; November 30, 2010. Issued: June 18, 2013. US Patent # 7,902,350: Issue Date March 8, 2011. Method for

sheep, with some individual flocks with 10–15% losses attributable to OJD. A total of 7.6 million carcasses were examined (38%; 2.9 million from the HPA). Overall, 16% of slaughter consignments (sheep consigned to the abattoir from a single vendor) were positive for OJD, of which 94% were from the HPA. In the HPA, the percentage of animals with lesions attributable to OJD at slaughter fell progressively from 2.4% (10,406/432,860) at commencement of vaccination in 2000 to 0.8% (1,573/189,564) by 2009. Herd immunity from vaccination in the HPA was estimated at 70% by 2009, the target commonly espoused for an effective control program based on vaccination. This coincided with a progressive decrease in reports of clinical disease and mortalities in vaccinated flocks.

## Significance

We show a decrease in the prevalence of lesions attributable to OJD in NSW concomitant with initiation of voluntary vaccination, on-farm management plans, abattoir monitoring and feedback of animal prevalence data to sheep producers. We conclude that a target of $\leq 1\%$ regional prevalence of OJD affected sheep at slaughter is achievable using these interventions.

## Introduction (See also S1 Introduction in S1 File)

Johne's disease [1] is a chronic, incurable, fatal wasting disease of animals, particularly ruminant animals, caused by *Mycobacterium avium* subspecies *paratuberculosis* (MAP). Granulomatous enteritis, usually characterized by the presence of acid-fast staining MAP, is a consistent feature of the disease. Johne's disease occurs on all continents and infects domesticated [2–5] and wild animals [3–8]. For all practical purposes, sheep with OJD in Australia and New Zealand are infected by a single dominant S or "sheep" strain of MAP [9, 10]. The Telford strain, for which there is a whole genome sequence, is representative of this endemic strain [11]. The Australian S strain is slow growing and fastidious in culture requirements on artificial media compared with the "cattle" or C strain isolated from bovine Johne's disease (BJD) [5, 10]. While the C strain was cultured in Britain as early as 1912 [12], in contrast, reliable isolation of the S strain was not reported until the late-1990's, with cultures requiring incubation for up to 16 weeks [13]. Bovine Johne's disease is characterized by intractable diarrhea, leading to wasting and death. It is a major concern for the dairy industry worldwide, with disease control programs in many countries. In the USA for example, in 1997 when dairy herd MAP infection herd-level prevalence was 22% [14], the estimated cost of BJD was in excess of US$200 million /yr. By 2007 the USA herd-level prevalence had increased to 68% of dairy herds (95% of herds with >500 cattle) [2], with a probable commensurate increase in cost. Additionally, there is increasing concern among some scientists and medical practitioners that MAP may be zoonotic [4, 15, 16] and a potential factor in the pathogenesis of Crohn's disease in man [3, 4, 16–25].

While diarrhea is not a feature of OJD, the slowly progressive granulomatous enteritis leads to malabsorption of nutrients and chronic weight loss, before the animal succumbs 2–6 months later. In some flocks, where lambs are exposed to heavy contamination, deaths may be evident as early as 2 years of age. More often sheep succumb at 3–5 years of age [26]. Spread of infection is by the feco-oral route, with late-stage animals excreting as many as $10^8$ organisms/ gm of feces [27].

Monitoring the Efficacy of a Mycobacterium Avium Subspecies Paratuberculosis Therapy. US Patent # 8,507,251: Issue Date August 13, 2013 "Medium and Method for Culturing Mycobacterium Avium Subspecies Paratuberculosis" Serial No: 12/ 892,039 Filing Date September 28, 2010. This does not alter any of our adherence to all the PLoS ONE policies on sharing data and materials. There are no other patents, products in development or marketed products associated with this research to declare. IL, LD, ME, and LK have no competing interests to declare.

**Abbreviations:** AFO, Acid-fast organism; BJD, Bovine Johne's disease (paratuberculosis); "C" Strain, "Cattle" strain of MAP; DPI, Department of Primary Industries; DV, District Veterinarian; H&E, Haematoxylin and Eosin stain; HPA, High Prevalence Area; INC, Inconclusive; LPA, Low Prevalence Area; MPA, Medium Prevalence Area; MAP, Mycobacterium avium subsp. paratuberculosis; NEG, Negative; NLIS, National Livestock Identification System; NSW, New South Wales; OJD, Ovine Johne's disease; PIC, Property Identification Code; PDMP, Property Disease Management Plan; PFC, Pooled Fecal Culture; POS, Positive; RLPB, Rural Lands Protection Board; "S" Strain, "Sheep" strain of MAP; ZN, Ziehl-Neelsen stain.

Control of Johne's disease in infected herds or flocks is difficult [28], although culling of heavily infected animals or physical separation of animals with less severe infection may be partially effective [29]. In addition, vaccination against MAP may ameliorate disease [30–37] and be cost effective in combating Johne's disease [38, 39]. By delaying the onset and reducing the level of fecal shedding of MAP by infected sheep [34], vaccination may reduce the spread of infection within and between flocks [40].

The region of greatest concern for OJD at the initiation of the study in 1999 was South Eastern Australia (see map, Fig 1), where OJD detections had been increasing exponentially in the state of NSW in the two decades since first detected in 1980 [41]. In 1998, for example, 204 flocks were diagnosed with OJD, when the cumulative total of infected flocks was 441 [42–44].

In the other Australian states, disease had not been detected in Victoria until 1989 in a single ewe introduced from Tasmania and was not confirmed in home-bred sheep until 1995 [45]. Infection was confirmed on mainland South Australia in 1997 while the first detection in Western Australia occurred in 2000 [43, 46].

A regulatory control program, based on restricting the movement of infected sheep, had commenced in NSW in 1996 [47] and this program was later expanded to a national program in 1998 in response to concern about disease spread to other states [44].

Detection initially relied on costly on-farm sampling, using serology with confirmatory histopathology on reactors or clinically affected animals. Serology was complemented with, then progressively superseded by, the more sensitive and cost effective Pooled Fecal Culture (PFC) in NSW from November 1999 [48]. The introduction of the PFC also overcame the potential liability of vaccination interfering with serological testing.

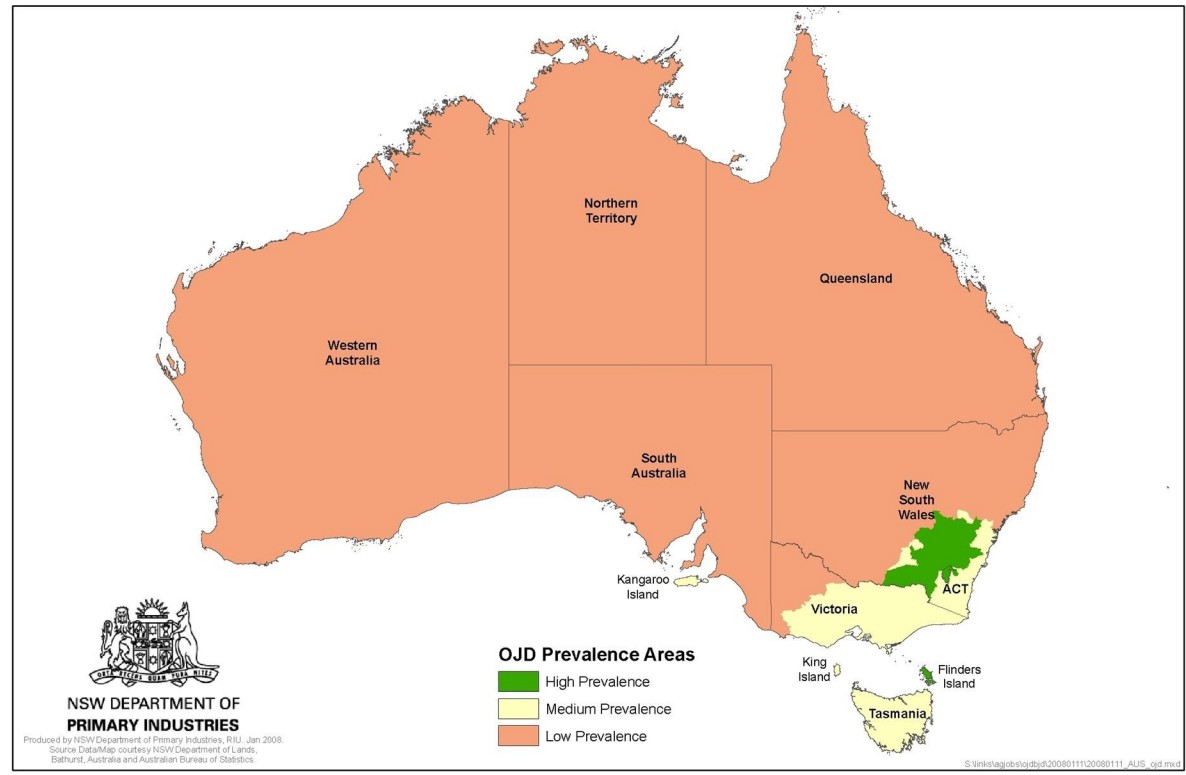

**Fig 1. Map of Australia showing regions of maximal OJD concern–prevalence Areas implemented 31 March 2008.** (Reprinted under a CC BY license, with permission from NSW DPI, original copyright 2008).

Beginning with the National Control and Evaluation Program in 1998 [49, 50], there has been a concerted attempt to evaluate the cost of OJD in Australia [51]. The sheep industry is very important to Australia, contributing as much as AU$4.3 billion in terms of direct Gross Value of Production. In 2008–09, the total number of sheep in Australia was estimated at 72 million, the gross value of sheep and lamb slaughtered was AU$2.5 billion and the gross value of wool was AU$1.8 billion. The most important states for sheep and wool are NSW, Victoria and Western Australia with NSW contributing approximately 21% of the gross national value of sheep meat and 34% of the value of wool [51].

An understanding of the complex factors in disease spread evolved over time. Straying sheep, use of common facilities including roads or yards, or local contamination of water run-off frequently led to progressive, but usually slow, spread to neighboring flocks within sub-catchments [52, 53]. In contrast, spread to new regions was due to the unwitting sale or movement of infected sheep, usually sub-clinically affected [42, 52]. There was no evidence of significant spread along major rivers ([42], by native macropods (kangaroos and wallabies) [7, 54] or rabbits [54].

Abbott & Whittington (2003) discuss in detail factors contributing to the spread of OJD and the differing environmental factors that influence potential for spread over an extended timeframe (30 years) in the HPA, MPA and LPA [55].

Windsor & Whittington (2020) in a recent review of the achievements of the National OJD Control Program in Australia over the past 2 decades acknowledge the contribution of abattoir monitoring and vaccination in dramatically reducing the impact of OJD on the sheep industry and rural communities. They recommend a renewal of the program to encourage infected flocks to continue vaccination to prevent recrudescence and minimize the ongoing risk of spread to clean flocks and low-risk areas [56].

Vaccination was first suggested as a likely cost effective strategy for controlling OJD in NSW in 1999 [57]. Subsequently, a killed whole-cell vaccine for OJD ("Gudair" ® - CZ Veterinaria, Porriño, Pontevedra, Spain) containing bovine strain 316F was approved for use under a special government permit but restricted to heavily infected flocks in NSW. Vaccination by sheep owners commenced on a voluntary basis in January 2000, in conjunction with on-farm Property Disease Management Plans (PDMPs) [50, 58]. The vaccine was subsequently registered for use in all states in April 2002. This followed release of preliminary results in controlled trials confirming the efficacy of vaccination in three heavily infected merino flocks in NSW [34]. Vaccination reduced both mortalities and the number of animals shedding MAP in their feces by 90%, as well as reducing the overall level of bacterial excretion by more than 90%.

Wide-scale monitoring for OJD of adult sheep consigned to NSW abattoirs commenced in late 1999 (this study, [53, 58–61]). The introduction of abattoir monitoring complemented existing on-farm testing. It helped determine the distribution and prevalence of this rapidly spreading disease and enabled more precise setting of prevalence area boundaries. Initially, only confirmed positive consignments were reported to the individual sheep producer [58]. However, the progressive introduction of the National Livestock Identification System (NLIS) [53], with markedly improved traceability to property-of-origin, enabled the results for both OJD positive and negative consignments to be reported to producers from January 2003.

This observational report documents the progressive changes in OJD prevalence following the sale of 12,021,963 vaccine doses and 7,635,513 carcass examinations in NSW, Australia from 1999 to 2009. During this period the sheep population of NSW decreased by 50% from approximately 35.4 million in 2000 to 17.7 million in 2009 [62–64].

## Methods

All activities and investigations were conducted in accordance with the NSW Agriculture OJD Policy Manuals issued in August 1999 and November 2001 [65, 66] and the National Johne's Disease Standard Definitions and Rules for Sheep [67–70]. Where appropriate we have defined the study parameters in accordance with established consensus-based reporting standards for diagnostic test accuracy studies for paratuberculosis in ruminants [71].

### Animal ethics approval

Animal ethics approval was not required as this case study did not involve any intervention with live animals. The research procedures described in the paper did not commence until after the animals had been killed in the normal course of the meat processing and livestock slaughter system, which is subject to separate national and NSW statutory animal welfare standards.

### Prevalence areas

OJD Prevalence Areas or Zones across Australia were initially established on 1$^{st}$ July 1999 using the results of on-farm surveillance, following tracing of sheep movements and investigation of clinical disease [58]. With the introduction of abattoir surveillance in November 1999, monitoring became more comprehensive with the boundaries of the prevalence areas progressively adjusted as OJD continued to spread and prevalence changed over time [53, 59]. For the purposes of standardization, we use the "High" (HPA), "Medium" (MPA) and "Low" (LPA) prevalence areas as implemented nationally on March 31, 2008 (Fig 1) [72]. Hence, irrespective of the date of monitoring or vaccination, the analysis of individual property data was based on the prevalence area status in March 2008.

### Property identification and tracing

The NLIS for sheep is based on a unique Property Identification Code (PIC) that identifies the state, local government region (RLPB–Rural Lands Protection Board) and property of origin of sheep consigned for slaughter. The NLIS was progressively implemented from 2002 to 2009. By July 2007, all sheep were required to be permanently ear-tagged with the PIC identifier for the property of birth. Sheep sent for slaughter were required to be accompanied by a National Vendor Declaration which recorded details of the owner and the property from which they were consigned (including the PIC from January 2006) [59].

Introduction of the NLIS greatly facilitated tracing, recording of vaccine sales, identification of consignments at slaughter, allocation to a prevalence area and reporting of abattoir monitoring results to producers. When the PIC was unknown, monitored consignments were allocated to a prevalence area based on Locality of Origin reported by the abattoir.

### Vaccine

The Spanish whole-cell oil adjuvant vaccine for OJD ("Gudair" ®), containing 2.5mg/ml of killed bovine MAP strain 316F as the sole active ingredient, is the only OJD vaccine approved for use in Australia. A single dose is considered to provide protection for the life of the animal (normally 5–6 years). Data on the number of doses of Gudair® vaccine used were obtained from two complementary sources: records of the number of vaccine doses used on each sheep property maintained for NSW Department of Primary Industries (DPI) by individual RLPBs, and vaccine sales data provided by the vaccine distributor (initially—CSL Limited, Parkville, Melbourne, subsequently Pfizer Animal Health, Sydney, Australia, now Zoetis Australia). Vaccine sales data were consolidated into a Microsoft Access database for analysis.

**Vaccine regulation (See also S2 Vaccine regulation in S1 File).**   Vaccine was initially approved in January 2000 for use on up to 50 individual properties in the NSW HPA with confirmed unacceptable losses attributable to OJD, i.e. ≥5% annual mortality. This was extended in July 2001 to 150 properties. It was approved for wider use nationally in April 2002, although still subject to restrictions. Unrestricted access to vaccine for any flock in NSW was approved in July 2003 [34, 53, 58, 66, 73].

**Vaccination procedures (See also S3 Vaccine procedures in S1 File).**   Vaccine was sold in 100ml or 250ml packs with each sheep receiving a 1ml dose subcutaneously in the upper neck behind the ear using an automatic syringe.

Standard Operating Procedures were developed to minimize the risk of operator self-inoculation and a national register of accidental self-inoculation was maintained by Pfizer Animal Health. To optimize the advantage of vaccination before significant exposure to MAP, it was recommended as a single dose in lambs from 4–16 weeks of age in NSW. Vaccination of older sheep was initially only permitted with approval of NSW DPI. Some producers vaccinated older lambs, hoggets (12–24 months-old) and/or 2- year-old ewes, while whole of flock vaccination was undertaken in a number of infected flocks.

## Property Disease Management Plans (PDMPs), financial assistance and producer workshops (See also S4 Property Disease Management Plans (PDMPs), financial assistance and producer workshops in S1 File)

The concept of formal approved PDMPs was promoted through a series of workshops, commencing in February 2002, with vaccination as a key control measure complemented by a range of other management strategies. A further series of workshops was held within the HPA and MPA between July and August 2003 [69, 74].

In order to implement PDMPs, financial assistance was made available to those producers whose flocks were confirmed infected with OJD prior to November 2002. Most of this funding was utilized for purchase of vaccine in 2002 and 2003.

## Abattoir monitoring

Monitoring was conducted from November 1999 to December 2009 in 18 abattoirs slaughtering sheep from NSW. The abattoirs were located respectively in:

- NSW (11)–Export abattoirs at Deniliquin, Dubbo, Goulburn and Wallangarra. Domestic abattoirs at Cootamundra, Cowra, Gundagai, Junee, Mudgee, West Wyalong and Young.

- Victoria (4)–Ararat, Brooklyn (Melbourne), Cranbourne and Warrnambool.

- South Australia (3)–Lobethal, Murray Bridge and Port Pirie.

Results were analyzed based on the calendar year of monitoring. To ensure comprehensive regional coverage in areas with differing seasonal sheep turnoff patterns, monitoring in the four major export abattoirs in NSW was maintained where possible throughout the year. These abattoirs process approximately 70% of all adult sheep slaughtered in NSW annually, with up to 10 sheep slaughtered per minute. Seasonal closure of abattoirs or other constraints meant that abattoirs were not able to be monitored continuously during the study period. The remaining 14 domestic and interstate abattoirs were monitored on a rotational basis depending on throughput of eligible age sheep.

**Direct consignments.**   Direct consignments of sheep ≥ 2-years-old, which were sent for slaughter from flocks in NSW as part of routine animal production turn-off, were screened for the presence of OJD using methods previously reported in detail [75–77]. Inspectors aimed to

monitor 90% of sheep in all eligible direct consignments on all slaughter shifts monitored. **Definition:** *Direct Consignment*—derived from a single vendor/PIC/property either consigned direct to the abattoir or through a saleyard without mixing with other consignments.

### Inspection, sampling & data collection (See also S5 Inspection, sampling & data collection in S1 File)

In brief, all inspections were undertaken by qualified meat inspectors approved to monitor for OJD after completing a national training workshop. These were conducted in an abattoir that slaughtered consignments from heavily infected properties. The monitoring procedures were detailed in a comprehensive training manual [70, 78–80].

Inspectors examined visually, and by palpation, the terminal ileum and adjacent lymph nodes for changes, including thickening of the intestinal wall, enlarged lymph nodes, lymphatic cording or mesenteric edema (Figs 2–4). The inspectors performed this task at a rate of up to 10 animals/minute. The animal-level sensitivity and specificity were estimated at 70% and 97% respectively following a research trial involving 5 consignments (1131 sheep) from known high prevalence infected properties [77].

Animals with suspect lesions were sampled (up to a maximum of three animals from each consignment–see below "Histopathology"). Where more than three suspect sheep per consignment were observed, inspectors could accumulate specimens and submit the most severely affected viscera after the whole line had been inspected. Sampling of sheep was therefore non-random and was biased towards those with the most obvious gross pathology. The animal-level and flock-level specificity of histopathology were both assumed to be 100% [55].

Data collected by the inspectors for each consignment included: Abattoir; Date killed; Consignment reference (abattoir Lot number); Number of animals killed and inspected; Owner of the sheep; Area/Locality of origin; NLIS PIC (where available); Age of the animals (minimum 2 years or older) and the identity of the Inspector (NSW abattoirs only).

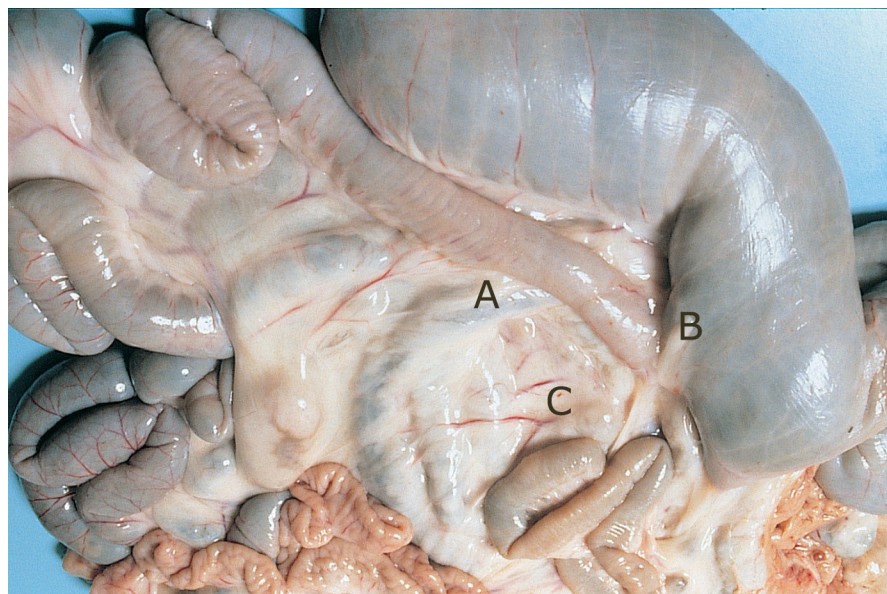

**Fig 2. Ileo-cecal area of an OJD affected sheep showing histopathology sampling sites.** A: ileo-cecal valve, B: thickened terminal ileum and C: enlarged ileo-cecal lymph nodes.

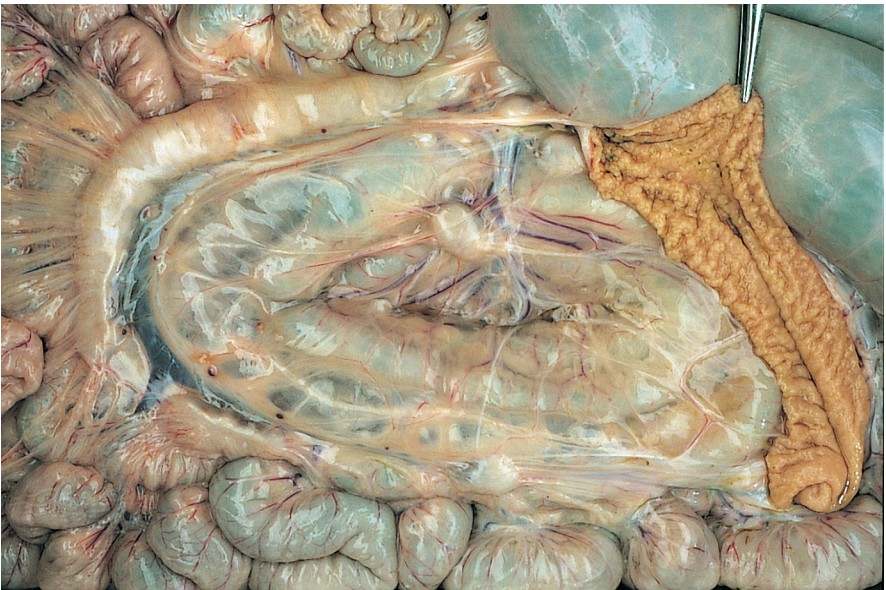

**Fig 3. Intestinal tract of an OJD affected sheep.** Note the thickened ileum, enlarged lymph nodes, prominent lymphatic vessels and edema of the mesentery, and the raised ridges (rugae) on the mucosal surface of the opened terminal ileum.

The number and percentage of individual sheep with gross lesions resembling OJD ("total lesions uncorrected") and the number of sheep sampled for histopathology were recorded for each consignment. The percentage of gross lesions was calculated from the number of sheep carcasses inspected, not the number killed. The percentage of animals with "lesions attributable to OJD" was determined pro-rata based on the results of histopathology on the animals sampled (see below "Correction of Data for Lesions Attributable to OJD").

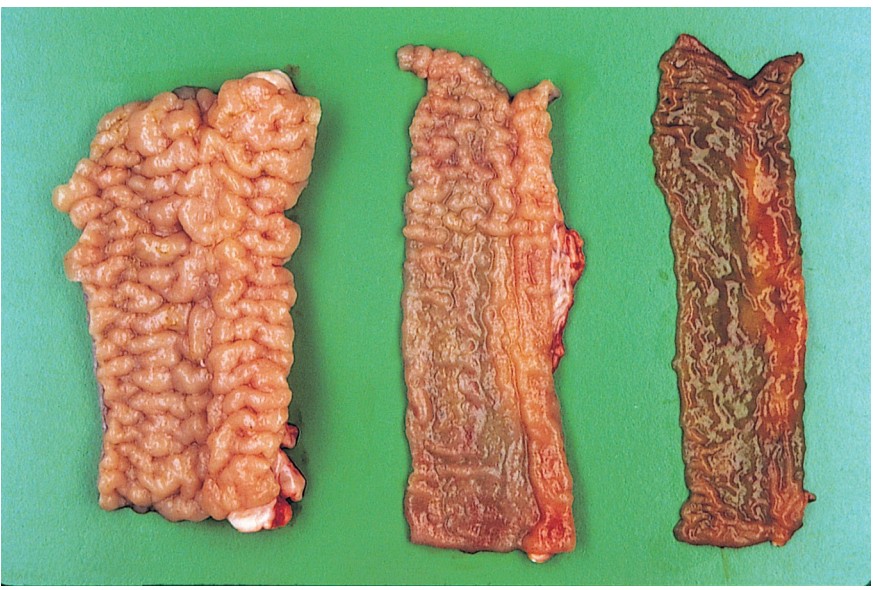

**Fig 4. Intestinal mucosa of an OJD affected sheep.** Note the prominent mucosal rugae in the severely affected section of the terminal ileum (left), the segmental nature of the ileal lesion (center) and the relatively normal mucosal lining of the jejunum (right).

Data were recorded in a standardized format on inspection sheets and transmitted daily by facsimile to a central facility at Orange (1999–2003), then Wagga Wagga NSW (2003–2009). Inspection details and laboratory results were entered into a Microsoft Access database maintained by NSW DPI.

**Histopathology.** Histopathology is considered the "Gold Standard" diagnostic test for OJD. Sensitivity, however, was not assumed to be 100% because of the possibility that sites selected for histological examination were not representative of the sites observed with gross lesions [55]. Initially, samples from the terminal ileum, ileo-cecal valve and adjacent mesenteric lymph nodes were submitted for confirmatory histopathology (Fig 2).

From 2002, however, only the terminal ileum was examined after a formal research trial demonstrated a minimal loss of sensitivity with this modification to the procedure (unpublished report to Sub-Committee on Animal Health Laboratory Standards—Sept. 2002). This change to methodology permitted the terminal ileum from three carcasses to be examined on a single microscope slide, thereby reducing laboratory costs by up to two thirds.

The suspect tissues were processed using standard procedures and the sections were stained by H&E (Hematoxylin and Eosin) and Ziehl-Neelsen (ZN) for histopathogical examination and determining the presence of acid-fast mycobacteria [81, 82].

**Case definition.** Throughout the term of the project the following case definition was employed: "An OJD positive animal was defined as one with gross pathology at slaughter, confirmed by histopathology as showing granulomatous inflammation of the terminal ileum typical of OJD and presence of acid-fast organisms (AFO's) in the specific animal or another animal in the same consignment."

**Classification of samples and consignments.** Samples were initially classified on the basis of histopathology (Fig 5A–5F) as:

- "Positive (Multibacillary) Histopathology" **(POS)**–presence of granulomatous changes typical of OJD with acid-fast organisms (AFOs) detected (Fig 5A–5C), or

- "Inconclusive (Paucibacillary) Histopathology" **(INC)**—presence of granulomatous changes resembling OJD but no AFOs detected [83] (Fig 5D–5F), or

- "Negative Histopathology" **(NEG)**–absence of granulomatous changes resembling OJD and no AFOs detected.

Samples identified initially as "INC" were reclassified as "POS" if other samples from the same consignment were "POS". For the purpose of retrospective data analysis, remaining "INC" animals were reclassified as "POS" if OJD had been confirmed in the flock prior to monitoring or within the subsequent two years [83].

Consignments with one or more "POS" samples were classified as OJD "Positive". Consignments which remained "Inconclusive" after retrospective review of flock history were removed from the dataset. The remaining consignments with no lesions or "NEG" histopathology were classified as OJD "Negative".

**Correction of data for lesions attributable to OJD (See also S5 Inspection, sampling & data collection in S1 File).** A range of conditions, such as coccidiosis, parasitism, salmonellosis, and yersiniosis, can cause lesions which grossly resemble OJD but yield OJD "NEG" histopathology. Hence, the number of sheep with lesions truly attributable to OJD in "Positive" consignments was required to be estimated. This was achieved by pro-rata correction (based on the results of histopathology) of the number of animals with lesions grossly resembling OJD as originally reported by the inspector (see S5 Inspection, sampling & data collection in S1 File for examples).

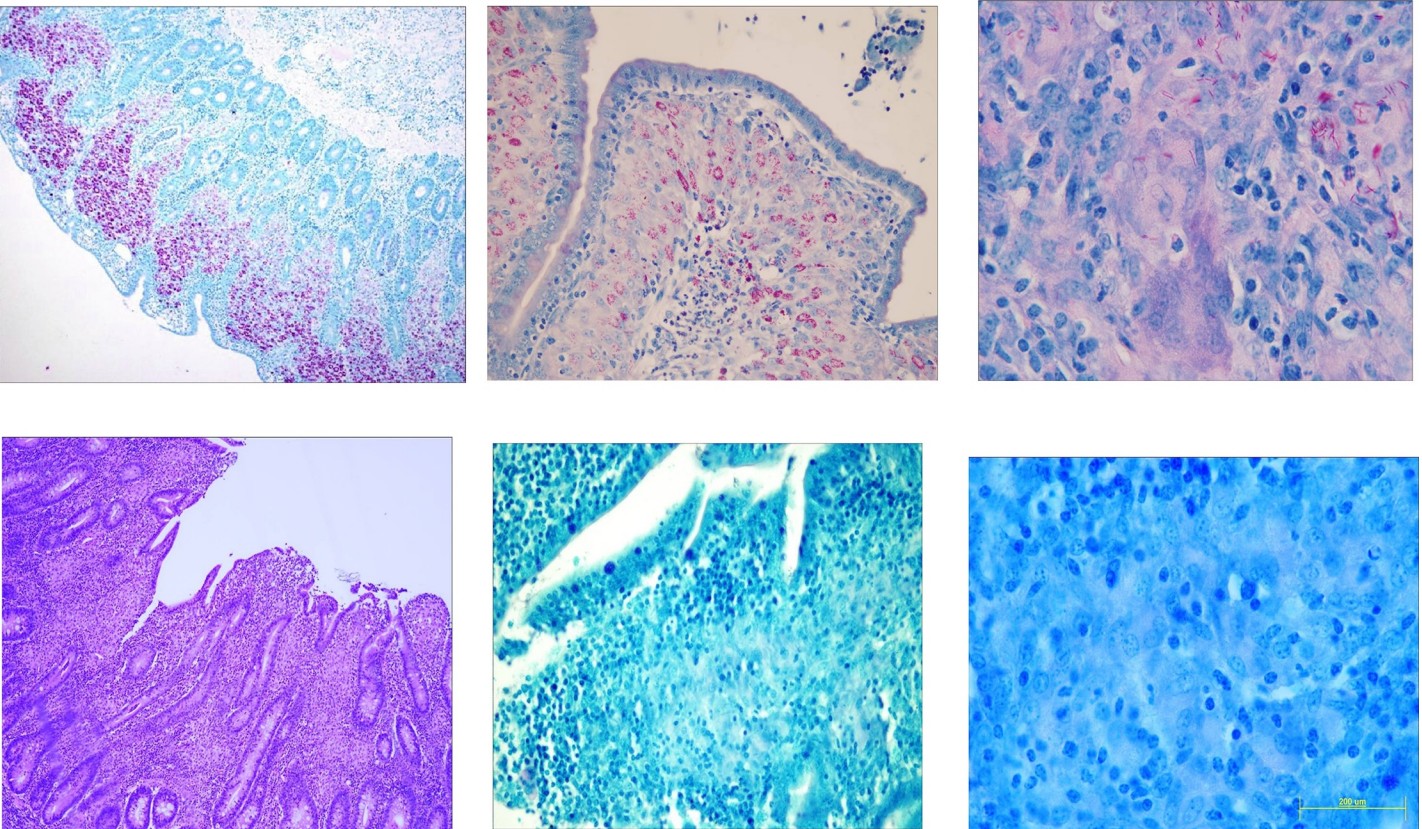

**Fig 5.** A-F. Histopathological changes due to OJD in the lamina propria of the terminal ileum. A) Multibacillary–masses of red staining acid-fast organisms (AFOs) typical of MAP bacilli (ZN X40); B) Multibacillary–aggregations of AFOs in the cytoplasm of epithelioid cells (ZN X400); C) Multibacillary–large numbers of acid-fast bacilli (MAP) in the cytoplasm of epithelioid cells and interstitial connective tissue (ZN X1000); D) Paucibacillary—showing granulomatous enteritis (H&E X100); E) & F) Paucibacillary–granulomatous enteritis but no AFOs evident (ZN X400 & X1000).

The percentage of OJD-positive sheep in a consignment or prevalence area was estimated from the **"corrected"** number of positive lesions compared with the total number of sheep inspected for that consignment or period.

**"Corrected"** data were utilized for all analyses.

**Specificity of visual inspection.** The specificity of visual inspection for OJD was determined by reviewing the histopathological findings on lesions submitted to the laboratory. The relative frequency of multibacillary and paucibacillary pathology was also examined.

**Feedback to producers (See also S6 Feedback to producers in S1 File).** OJD positive results were reported to the relevant RLPB District Veterinarian (DV) [58]. The DV in turn personally notified the owners of the sheep. Where the producer identified by tracing expressed doubt regarding the identity of their positive consignment, a risk assessment, and on-farm testing if required, was conducted to confirm or allay suspicion. Producers were notified directly by NSW DPI of all "Negative" consignments monitored after January 2003.

Reports to producers and DVs utilized a mail-merge procedure in Microsoft Word linked to the NSW PIC register in the Microsoft Access Abattoir Monitoring database.

## Flock and sheep numbers

Annual adult sheep and flock numbers (containing ≥50 adult sheep) in NSW (2000–2009) were obtained from the RLPB (later Livestock Health and Pest Authority) Annual Land &

Stock Returns [62–64]. Flocks were allocated to prevalence areas based either on RLPB or by property locality in the case of RLPBs which were split by prevalence area boundaries.

**Age at vaccination and cumulative (3 year) percentage of adult sheep vaccinated.**   The age categories vaccinated and number of doses were recorded for 2000–2002 and 2007–2009. For the purposes of analysis, data for 2003–2006 were assumed to be similar to 2007–2009.

The vaccine doses sold annually as a percentage of adult sheep provide a raw estimate of annual vaccination usage. However, the vaccine is "one dose for life" and sheep may be vaccinated as lambs (<12-months-old, but recommended at <4-months-old), hoggets (12-24-months-old) or adults (≥ 2-years-old). Hence, the percentage of adult sheep currently vaccinated was determined by the cumulative number of adults remaining in the flock that were vaccinated over the preceding years.

For the purposes of estimating the "cumulative" (rolling 3 year) percentage of vaccinated adult sheep (corrected for age at vaccination), it was assumed that: a) all vaccinated sheep were culled when they were nearing 5 years-old (representing 3 years of productive adult life); b) those vaccinated as lambs and hoggets contributed to the vaccinated adult numbers when 2 years-old, c) all sheep vaccinated as adults were vaccinated when they were 2 years-old and d) lambs and hoggets destined for slaughter at <2 years of age were not vaccinated [84]. Vaccine orders per flock per year were determined from 2000–2008 (RLPB records) and 2007–2012 (Pfizer Database).

## Sheep breeds affected

**On-farm investigations 1982–2006.**   The sheep breed affected was determined for 779 NSW DPI laboratory submissions confirmed positive for OJD following on-farm sampling from 1982 to 2006.

**Abattoir monitoring NVDs 2003–2007.**   In the present study, a subset of 2,249 National Vendor Declarations from 2003–2007 were identified where the sheep breed was recorded as Merino or Crossbred.

## Generation of graphs

All graphs were generated from data in Excel spreadsheets (Microsoft) or Prism (GraphPad Software, Inc.).

## Results

### Prevalence areas

There was a progressive increase in the detection of OJD infected flocks in NSW starting in 1980, with an exponential increase in detection over the next two decades [42, 44]. Fig 1 shows that the highest prevalence of OJD at 31 March 2008 within Australia was in NSW, Victoria, Flinders Island (Tasmania) and Kangaroo Island (South Australia).

Fig 6 shows the location within NSW of the areas designated as being of High, Medium and Low prevalence for OJD. The NSW prevalence area boundaries were based on the regional prevalence and location of individual infected flocks determined by abattoir monitoring from 1999–2007. The results of abattoir monitoring from 1999–2009 (32,032 consignments) are represented in Fig 6.

### Abattoir monitoring

Of the 32,210 consignments monitored from 1999 to 2009, 178 (0.6%) "Inconclusive" consignments were removed from the dataset as their OJD status was unable to be resolved– 1.2%

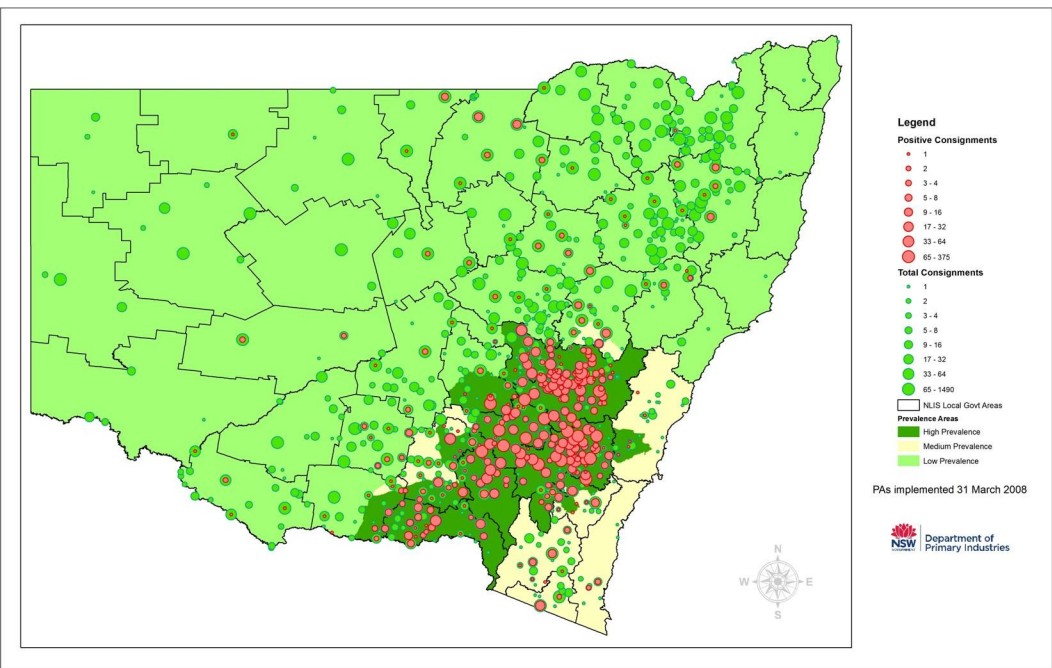

**Fig 6. Map of NSW showing number of abattoir consignments monitored by locality and OJD prevalence area.** Shown are the total number of consignments (green disc) and numbers of "Positive" consignments (red disc) for each locality monitored from 1999–2009 (32,032 consignments). The size of the disc is indicative of the number of consignments monitored. Regions of OJD prevalence are denoted by color-coding. High Prevalence Area (HPA) is shown in dark green, Medium (MPA) in light cream and Low (LPA) in light green. (Reprinted under a CC BY license, with permission from NSW DPI, original copyright 2010).

(156/13,122) of consignments from the HPA, 0.3% (5/1560) of consignments from the MPA and 0.1% (17/17,528) of consignments from the LPA.

The remaining 32,032 consignments classed as "Positive" or "Negative" for OJD were included in the analysis, of which 41% were from the HPA (12,966/32,032—Table 1, Fig 6).

**Table 1. Number of sheep consignments monitored at Abattoirs in NSW from 1999–2009 by prevalence area.**

| Year | High | Medium | Low | Total/year |
|---|---|---|---|---|
| 1999 | 148 | 12 | 68 | 228 |
| 2000 | 2,113 | 287 | 2,291 | 4,691 |
| 2001 | 1,617 | 204 | 1,950 | 3,771 |
| 2002 | 1,820 | 162 | 1,735 | 3,717 |
| 2003 | 945 | 133 | 1,043 | 2,121 |
| 2004 | 1,119 | 100 | 1,263 | 2,482 |
| 2005 | 583 | 83 | 1,123 | 1,789 |
| 2006 | 1,570 | 126 | 1,988 | 3,684 |
| 2007 | 1,228 | 230 | 2,274 | 3,732 |
| 2008 | 1,005 | 165 | 1,974 | 3,144 |
| 2009 | 818 | 53 | 1,802 | 2,673 |
| Total 1999–2009 | 12,966 | 1,555 | 17,511 | 32,032 |

The total number of sheep consignments monitored at abattoirs in NSW from 1999–2009. Stratification is by OJD prevalence area. Total number per year (right-hand column) and cumulative number per Prevalence Area from 1999–2009 (bottom row).

**Table 2. Number of carcasses inspected in NSW (1999–2009) by prevalence area.**

| Year | High | Medium | Low | Total/Year |
|---|---|---|---|---|
| 1999 | 32,734 | 3,224 | 20,163 | 56,121 |
| 2000 | 432,860 | 42,234 | 596,883 | 1,071,977 |
| 2001 | 344,097 | 27,990 | 507,196 | 879,283 |
| 2002 | 386,822 | 28,736 | 418,611 | 834,169 |
| 2003 | 185,452 | 24,851 | 215,753 | 426,056 |
| 2004 | 243,962 | 23,382 | 291,424 | 558,768 |
| 2005 | 151,604 | 19,892 | 289,907 | 461,403 |
| 2006 | 444,272 | 28,744 | 504,446 | 977,462 |
| 2007 | 282,677 | 56,805 | 570,022 | 909,504 |
| 2008 | 238,126 | 38,444 | 502,410 | 778,980 |
| 2009 | 189,564 | 11,256 | 480,970 | 681,790 |
| Total 1999–2009 | 2,932,170 | 305,558 | 4,397,785 | 7,635,513 |

The numbers of sheep carcasses inspected from 1999–2009 stratified by NSW OJD Prevalence Area. Total number per year (right-hand column) and cumulative number per Prevalence Area from 1999–2009 (bottom row).

Of the 7,635,513 inspected carcasses, 38% (2,932,170) were from the HPA (Table 2). OJD was identified in 15.9% of **all** consignments (5,109/32,032—Tables 1 and 3), of which 94% were from the HPA (4,823 /5,109; Table 3). Within the HPA itself, 37% (4,823/12,966—Tables 1 and 3) of consignments were "Positive" for OJD.

Of the 1,379,819 carcasses which were inspected in the "Positive" consignments, 94% (1,300,022) originated from the HPA (Table 4). Using our established criteria (see Case Definition), 50,719 animals had OJD of which 96% (48,514) originated from the HPA (Table 5). Of the 1,300,022 carcasses in OJD "Positive" consignments which originated from the HPA, 3.7% (48,514) had OJD lesions (Tables 4 and 5).

During the course of the study, despite some variation between years, the percentage of consignments that were "Positive" in the HPA leveled off between 33% and 47% (Table 3 & Fig 7A) confirming that flock-level prevalence remained consistently high.

**Table 3. "Positive" consignments in NSW (1999–2009) by prevalence area.**

| Year | High | Medium | Low | Total/Year |
|---|---|---|---|---|
| 1999 | 49 | 2 | 2 | 53 |
| 2000 | 494 | 9 | 15 | 518 |
| 2001 | 551 | 12 | 18 | 581 |
| 2002 | 764 | 14 | 24 | 802 |
| 2003 | 368 | 14 | 4 | 386 |
| 2004 | 465 | 19 | 8 | 492 |
| 2005 | 210 | 4 | 11 | 225 |
| 2006 | 735 | 8 | 18 | 761 |
| 2007 | 525 | 21 | 18 | 564 |
| 2008 | 388 | 21 | 20 | 429 |
| 2009 | 274 | 10 | 14 | 298 |
| Total (+) | 4,823 | 134 | 152 | 5,109 |

The numbers of "Positive" consignments where OJD was detected from1999-2009 Stratified by NSW Prevalence Area. Total number per year (right-hand column) and cumulative number per Prevalence Area from 1999–2009 (bottom row).

**Table 4. Total number of sheep inspected in "positive" consignments (1999–2009) by prevalence area.**

| Year | High | Medium | Low | Total/Year |
|------|------|--------|-----|------------|
| 1999 | 13,846 | 495 | 549 | 14,890 |
| 2000 | 123,390 | 1,180 | 5,748 | 130,318 |
| 2001 | 123,673 | 3,005 | 4,419 | 131,097 |
| 2002 | 177,013 | 3,218 | 6,571 | 186,802 |
| 2003 | 85,571 | 3,374 | 1,635 | 90,580 |
| 2004 | 117,282 | 4,690 | 2,996 | 124,968 |
| 2005 | 66,416 | 1,300 | 2,373 | 70,089 |
| 2006 | 251,534 | 1,728 | 6,235 | 259,497 |
| 2007 | 150,898 | 6,397 | 4,577 | 161,872 |
| 2008 | 109,918 | 4,934 | 7,335 | 122,187 |
| 2009 | 80,481 | 2,449 | 4,589 | 87,519 |
| Total | 1,300,022 | 32,770 | 47,027 | 1,379,819 |

The numbers of sheep inspected in "Positive" consignments where OJD was detected from 1999–2009 stratified by NSW Prevalence Area. Total number per year (right-hand column) and cumulative number per Prevalence Area from 1999–2009 (bottom row).

Overall, the percentage of animals that were positive for OJD in **all** consignments originating from the HPA fell progressively from 2.4% (10,406/432,860) of the carcasses inspected at commencement of vaccination in 2000, to 0.8% (1,573/189,564) of the carcasses inspected in 2009 (Tables 2 and 5; Figs 7B and 8A).

However, during this period, the percentage of animals that were positive for OJD in **"Positive"** consignments from the HPA fell markedly from 8.5% (11,689/137,236) of the carcasses inspected in 1999–2000 to 2.0% (3,849/190,399) of the carcasses inspected in 2008–2009 (Tables 4 and 5; Fig 8B).

## Vaccination

Vaccination began in NSW in late 1999, initially in a major vaccination trial in three heavily infected self-replacing merino flocks [34] (i.e. closed flocks with no introductions). From

**Table 5. Total number of sheep with OJD lesions in "positive" consignments by prevalence area.**

| Year | High | Medium | Low | Total /year |
|------|------|--------|-----|-------------|
| 1999 | 1,283 | 26 | 1 | 1,310 |
| 2000 | 10,406 | 169 | 74 | 10,649 |
| 2001 | 6,197 | 228 | 78 | 6,503 |
| 2002 | 10,221 | 124 | 31 | 10,376 |
| 2003 | 4,057 | 215 | 181 | 4,453 |
| 2004 | 4,607 | 289 | 15 | 4,911 |
| 2005 | 1,505 | 15 | 57 | 1,577 |
| 2006 | 3,823 | 20 | 55 | 3,898 |
| 2007 | 2,566 | 89 | 41 | 2,696 |
| 2008 | 2,276 | 168 | 120 | 2,564 |
| 2009 | 1,573 | 139 | 70 | 1,782 |
| Total | 48,514 | 1,482 | 723 | 50,719 |

The numbers of sheep detected with OJD lesions in "Positive" consignments from 1999–2009. Stratification is by NSW Prevalence Area. Total number per year (right-hand column) and cumulative number per Prevalence Area from 1999–2009 (bottom row).

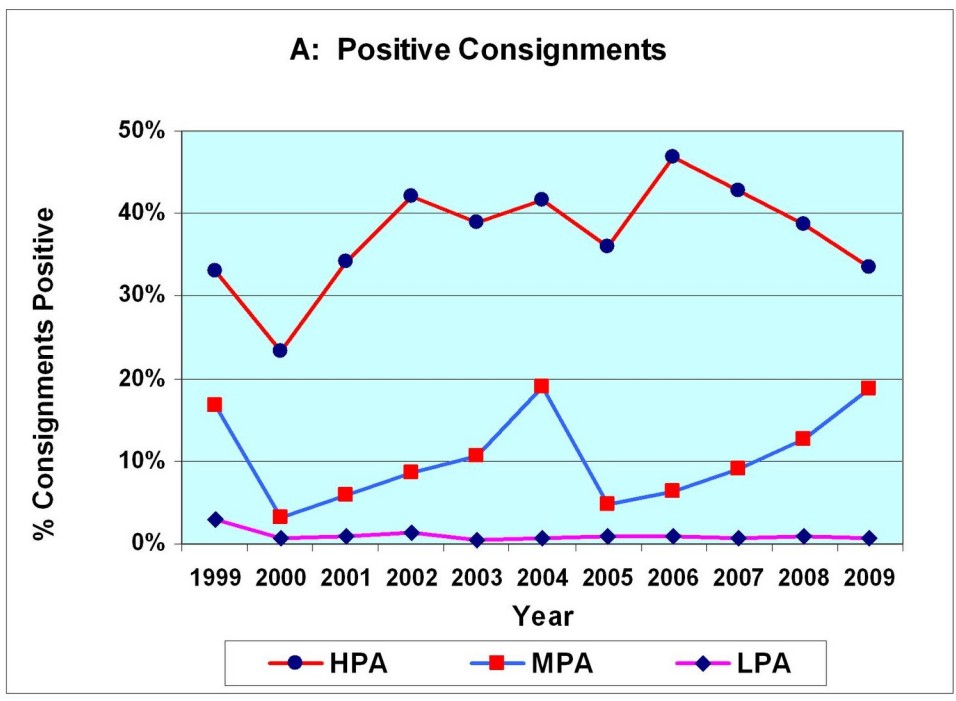

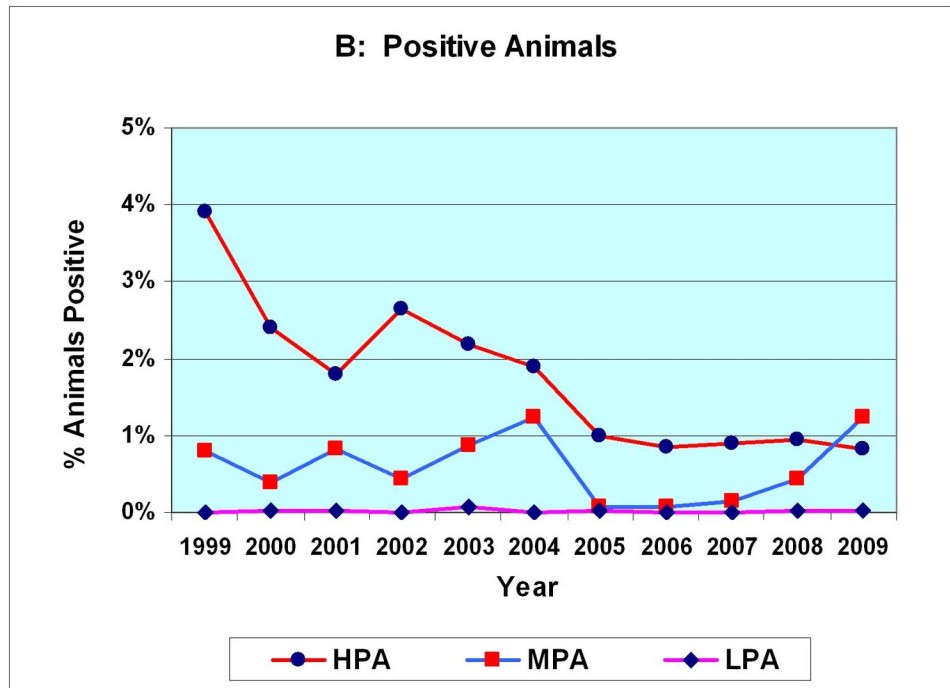

**Fig 7. Comparison of percentage of consignments or percentage of animals positive for OJD.** (A) Positive Consignments —the percentage of consignments that are positive remains relatively constant from 2001–2009 in the HPA. In the MPA there is an increase from 2005 to 2009. (B) Positive Animals—there is a progressive decrease in the numbers of positive animals as a percent of the total number of animals inspected in all consignments in the HPA. In contrast, there is a progressive increase in the percentage of positive animals from 2006 to 2009 in the MPA.

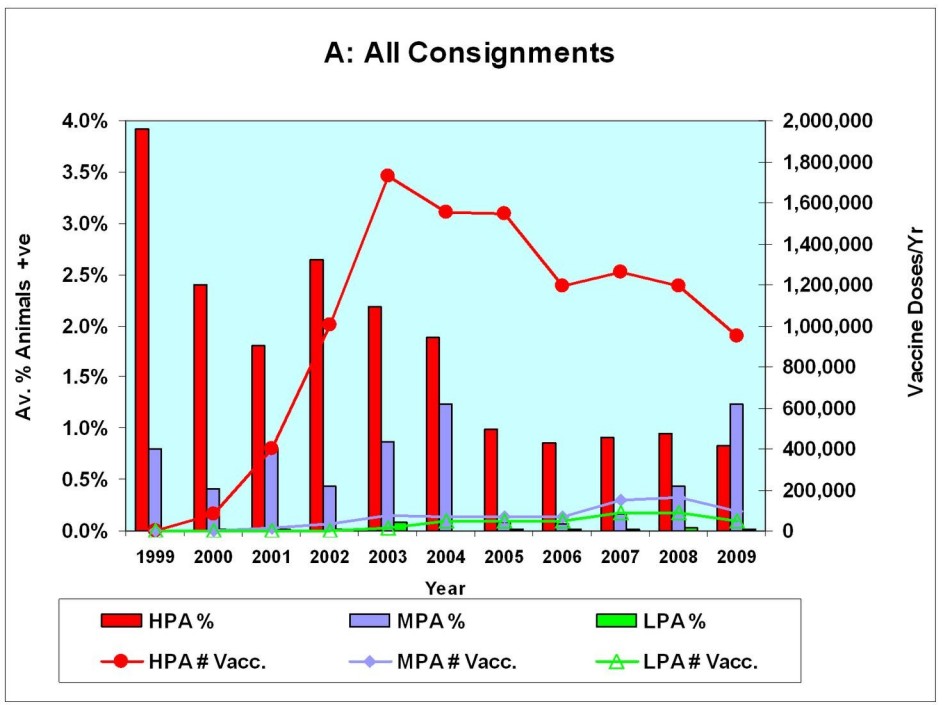

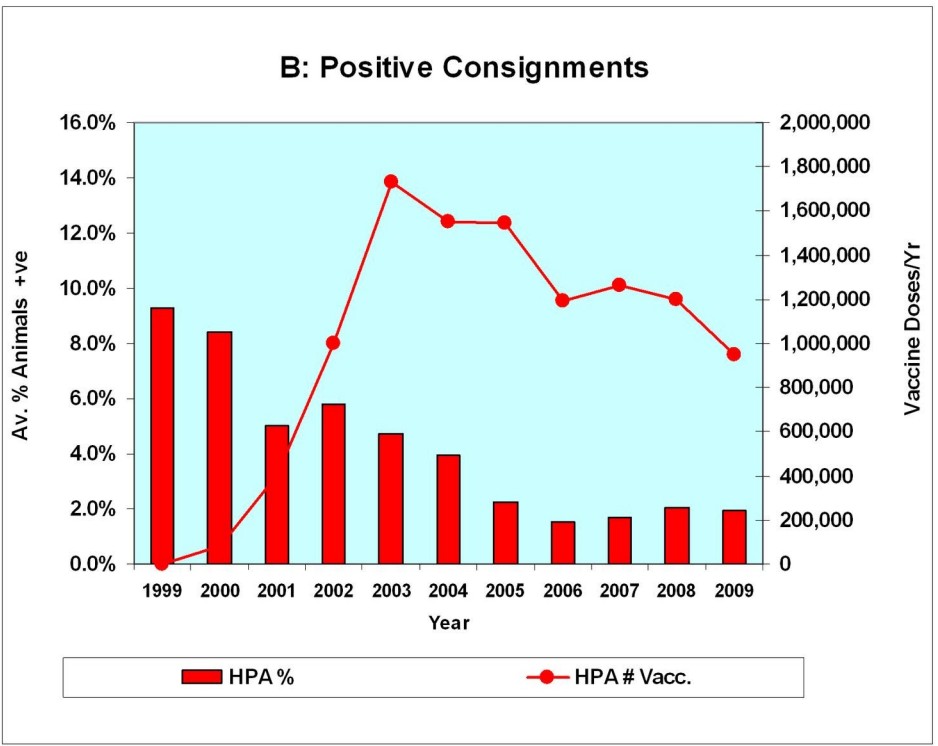

**Fig 8. Comparison of annual vaccine sales and percentage of OJD positive lesions (1999–2009).** Annual animal-level prevalence percentages are shown as vertical bars (scale on the left-hand vertical axis). Yearly sales of vaccine are shown by lines (scale on the right-hand vertical axis). (A) All Consignments (32,032)—Stratified by Prevalence Area. Note the progressive decline in animal-level prevalence in the HPA as vaccination progresses and the contrasting increase in animal-level prevalence in the MPA from 2006 to 2009. (B) Positive Consignments (4,823)—High Prevalence Area. Note the marked decline in % positive lesions from 8.5% in 1999–2000 to ~2% from 2006–2009 as vaccination progresses.

January 2000, vaccination was restricted to a limit of 50 heavily infected flocks whose annual mortality attributable to OJD was greater than 5%. A total of 63,000 doses had been ordered for 51 flocks by the end of September 2000. By April 2002, when the Gudair® OJD vaccine was approved and registered for use in all states of Australia and restrictions eased considerably, there were 237 heavily infected flocks (>5% annual mortality) enrolled in NSW allocated 574,150 doses.

Subsequently, vaccine use became more widespread with decreasing regulatory controls, particularly in the HPA. By 2009, a total of 12,021,963 doses had been administered, of which 91% were administered in the HPA (Table 6). In the HPA, the maximum number of vaccinations in any one year occurred in 2003 (1.73 million). Subsequently, there was a gradual but variable decline in the number of doses administered annually in the HPA, falling to 0.95 million doses by 2009.

In contrast, there was a progressive increase, albeit from a much lower base, in the number of vaccinations administered in the MPA, increasing 13-fold from 12,749 in 2001 to a maximum of 165,972 in 2008 (Table 6).

Given this decline in the animal-level prevalence of OJD in the HPA, the data were reviewed to determine whether there was any association between the use of vaccination and this trend in the regional animal-level prevalence of OJD. From the time that vaccination began in the HPA, there was a progressive decrease in the percentage of sheep with lesions per consignment (Fig 8A) and per "Positive" consignment (Fig 8B).

A more detailed analysis revealed a time-dependent positive correlation between the increasing number of vaccinated sheep (as vaccination of the HPA population proceeded) and a declining number of "Positive" consignments with a high percentage of infected animals (Fig 9A–9F). This decrease was most pronounced for OJD "Positive" consignments categorized with >10% of positive animals. There were 28.5% of consignments in this category in 2000 falling to ~2.3% of consignments from 2005–2009 (Fig 9A and 9B).

Furthermore, there was a progressive increase in consignments with a low percentage of infected animals as vaccination proceeded. Positive consignments with <1% of animals infected comprised ~34% in 2000 rising to ~50% by 2009 (See Fig 9C–9F for trends).

**Specificity of visual inspection assessed by histopathology.** Review of the histopathology results for 11,687 individual sheep in consignments derived from the HPA confirmed 9,570 as

**Table 6. Number of vaccine doses sold in NSW (1999–2009) by prevalence area.**

| Year | High | Medium | Low | Total /year |
|---|---|---|---|---|
| 1999[a] | - | - | - | - |
| 2000 | 83,700 | - | - | 83,700 |
| 2001 | 400,852 | 12,749 | - | 413,601 |
| 2002 | 1,001,122 | 30,929 | - | 1,032,051 |
| 2003 | 1,730,590 | 75,745 | 13,416 | 1,819,751 |
| 2004 | 1,553,554 | 69,891 | 45,916 | 1,669,361 |
| 2005 | 1,545,161 | 65,117 | 50,073 | 1,660,351 |
| 2006 | 1,191,433 | 67,198 | 44,818 | 1,303,449 |
| 2007 | 1,261,933 | 147,155 | 88,262 | 1,497,350 |
| 2008 | 1,195,955 | 165,972 | 86,872 | 1,448,799 |
| 2009 | 950,450 | 93,300 | 49,800 | 1,093,550 |
| Total | 10,914,750 | 728,056 | 379,157 | 12,021,963 |

The numbers of vaccine doses sold in NSW stratified by OJD Prevalence Area. Total number per year (right-hand column) and Cumulative number per Prevalence Area from 1999–2009 (bottom row). [a]No vaccine doses were sold in 1999.

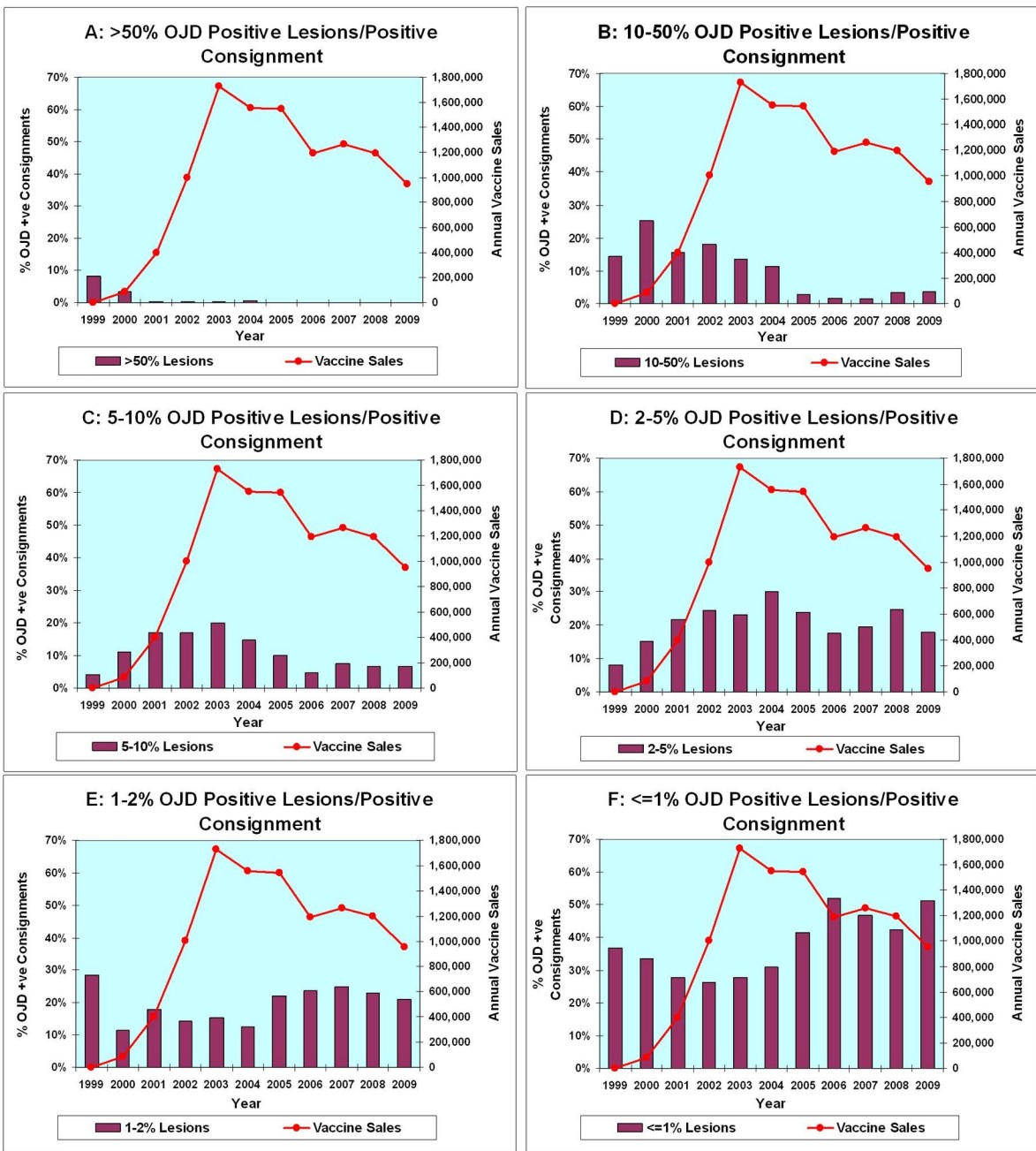

**Fig 9.** Percentage of lesions attributed to OJD in positive consignments from the HPA following institution of vaccination in 2000 (A-F). For clarity, the Results from the HPA only are displayed. A comparison of the annual number of vaccine doses sold with the percentage of positive animals per positive consignment. In each graph, the total doses of vaccine sold are repetitively represented by a line (scale on the right-hand side vertical axis) while the percentage of positive consignments in each percentile category is shown by vertical bars (scale on the left-hand side vertical axis). (A) shows data where the percentage of animals positive per positive consignment exceeds 50%. This only occurs in 1999 and 2000, when vaccine sales were negligible. (B) and (C) show a progressive fall in the percentage of consignments where 5%-50% of animals were positive per positive consignment. In contrast, by (F) (≤1%) an increasing percentage of consignments have a low percentage of animals positive as vaccination progresses. We interpret these results as indicating improvement in the control of OJD by vaccination.

OJD positive (either multibacillary or paucibacillary), an individual animal specificity of 82%. In the MPA the specificity was 63% (271/428) and in the LPA, where inspectors were deliberately encouraged to sample anything remotely suspicious, the specificity was 18% (87/1,028). The individual animal specificity was 76% for samples from the 32,032 consignments monitored from all prevalence areas in NSW.

Paucibacillary pathology attributable to OJD (typical histopathology but no acid-fast organisms) was detected in 17% (1,737/10,028) of the individual sheep sampled, while the remaining 83% were reported with the multibacillary form.

## Flock and sheep numbers

The number of flocks and sheep in NSW declined by approximately 50% from 31,895 flocks and 35.4 million sheep in 2000 to 16,419 flocks and 17.7 million sheep in 2009. This decline was primarily associated with an extended period of statewide drought from 2002–2009 (the "Millennium" drought).

The percentage of flocks and sheep in each prevalence area, however, remained relatively static over the decade (Table 7). The average flock size in NSW (2000–2009) was 1141 sheep, with an average of 282 sheep/flock in the HPA, 918 in the MPA and 1275 in the LPA.

**Age at vaccination and percentage of adult sheep vaccinated.** From 2000–2002, when vaccine use was restricted to the HPA, the age at vaccination was categorized as: 75% lambs (<12-months-old), 17% hoggets (12-24-months-old) and 8% adults (≥2-years-old). The lambs at time of vaccination were further sub-categorized as: 41% at 1–4 months-old (approved vaccinates), 11% at 5–6 months and 23% at 7–12 months.

In contrast, vaccine sales in the HPA from 2007–2009 were categorized as: 94% lambs (<12 months), 2% hoggets and 4% adults. Corresponding figures were 87%, 3% and 10% for sheep in the MPA (2007–2009) and 90%, 1% and 9% for sheep in the LPA.

The vaccine doses sold annually as a percentage of the adult sheep population by prevalence area and the estimated "cumulative" (rolling 3 year) percentage of vaccinated adult sheep (corrected for age at vaccination) is shown in Table 8. This provides an estimate of the level of "**herd immunity**" by prevalence area as vaccination progressed over time.

Annual vaccine sales in the HPA as a percentage of the adult sheep population rose rapidly (3.3% in 2001, 8.7% in 2002) before plateauing at ~16% from 2003–2009. In the MPA levels

**Table 7. Distribution of flocks and sheep in NSW (2000–2009) by prevalence area.**

| Year | No. of Flocks | | | | No. of Sheep | | | |
|------|-------|-------|--------|------|-------|------|--------|------|
| | Total | High | Medium | Low | Total | High | Medium | Low |
| 2000 | 31,895 | 37.1% | 11.5% | 51.4% | 35,411,370 | 34.1% | 8.4% | 57.4% |
| 2001 | 26,631 | 39.7% | 10.9% | 49.4% | 35,353,577 | 34.0% | 9.4% | 56.5% |
| 2002 | 25,277 | 40.6% | 11.1% | 48.3% | 33,096,199 | 34.9% | 10.1% | 55.0% |
| 2003 | NA | NA | NA | NA | 27,921,878 | 38.1% | 9.6% | 52.3% |
| 2004 | NA | NA | NA | NA | 27,065,066 | 36.3% | 8.9% | 54.9% |
| 2005 | 23,687 | 40.0% | 10.6% | 49.4% | 26,698,537 | 36.3% | 8.6% | 55.1% |
| 2006 | 24,111 | 40.3% | 11.5% | 48.2% | 26,233,654 | 35.8% | 9.0% | 55.3% |
| 2007 | 23,685 | 33.0% | 15.1% | 51.8% | 23,418,375 | 31.6% | 12.0% | 56.4% |
| 2008 | 20,670 | 38.8% | 12.3% | 48.9% | 21,453,983 | 34.6% | 9.8% | 55.7% |
| 2009 | 16,419 | 37.4% | 11.6% | 51.0% | 17,671,195 | 31.7% | 9.6% | 58.7% |
| Mean | 24,047 | 38.4% | 11.8% | 49.8% | 27,432,383 | 34.8% | 9.5% | 55.7% |

Distribution of flocks (≥50 adult sheep) and numbers of adult sheep by OJD prevalence area in NSW—2000 to 2009 (Data from RLPB records). Flocks were allocated to prevalence areas based on the boundaries imposed nationally on 31 March 2008. NA- not available.

**Table 8. Estimated % of sheep vaccinated in NSW (1999–2009) by prevalence area.**

| Year | Vaccine Doses (Annual) as % of Adult Sheep | | | | Cumulative (Rolling 3 Year) % of Adult Sheep Vaccinated | | | |
|------|------|--------|-----|-----|------|--------|-----|-----|
|      | High | Medium | Low | All | High | Medium | Low | All |
| 1999 | -    | -      | -   | -   | -    | -      | -   | -   |
| 2000 | 0.7% | 0.0%   | 0.0% | 0.2% | 0.0% | 0.0%  | 0.0% | 0.0% |
| 2001 | 3.3% | 0.4%   | 0.0% | 1.2% | 0.5% | 0.0%  | 0.0% | 0.2% |
| 2002 | 8.7% | 0.9%   | 0.0% | 3.1% | 2.2% | 0.1%  | 0.0% | 0.8% |
| 2003 | 16.3% | 2.8%  | 0.1% | 6.5% | 6.7% | 0.9%  | 0.0% | 2.6% |
| 2004 | 15.8% | 2.9%  | 0.3% | 6.2% | 16.2% | 2.5% | 0.0% | 6.1% |
| 2005 | 15.9% | 2.8%  | 0.3% | 6.2% | 32.1% | 5.7% | 0.2% | 12.2% |
| 2006 | 12.7% | 2.8%  | 0.3% | 5.0% | 44.7% | 7.6% | 0.5% | 16.9% |
| 2007 | 17.1% | 5.2%  | 0.7% | 6.4% | 64.5% | 7.3% | 0.9% | 21.8% |
| 2008 | 16.3% | 7.7%  | 0.7% | 6.8% | 57.9% | 10.1% | 1.3% | 21.6% |
| 2009 | 17.0% | 5.5%  | 0.5% | 6.2% | 70.8% | 17.8% | 1.8% | 25.2% |

The vaccine doses sold annually as % of adult sheep provides a raw estimate of regional vaccination level. However the vaccine is "one dose for life" and sheep may be vaccinated as lambs, hoggets or adults. Hence the % of adult flock currently of vaccinated status will comprise the cumulative number of adult sheep remaining in the flock that have been vaccinated over the preceding years (see Materials and Methods).

were ~3% from 2003–2006 before increasing to ~5–8% from 2007–2009. Vaccine usage remained very low in the LPA (~0.3–0.7% from 2004–2009). For the whole of NSW, vaccine usage was ~6% from 2003–2009.

In contrast the "cumulative" vaccination status in the HPA reached an estimated 16% by 2004, ~45% by 2006 and ~70% in 2009. The MPA exceeded 5% "cumulative" vaccination status by 2005, 10% by 2008 and 17% by 2009 while the LPA reached 0.9% "cumulative" vaccination status in 2007 and 1.8% in 2009. The "cumulative" vaccination status for the total sheep population of NSW was estimated to exceed 2.5% in 2003, 6% in 2004, 20% in 2007 and 25% in 2009.

A total of 1320 individual flocks/properties were vaccinated in the HPA from 2000–2008 (detailed records maintained by RLPBs) and 2483 flocks from 2007–2012 (Pfizer database)–see Table 9 and 10.

The number of flocks vaccinated annually in the HPA (Tables 9 and 10) increased progressively with the easing of restrictions on accessing vaccine. A total of 1320 individual flocks were vaccinated at least once between 2000 and 2008 (400–600 flocks per year from 2002 onwards) and 2483 flocks were vaccinated at least once from 2007–2012 (~1300 flocks

**Table 9. Number of flocks vaccinated in the high prevalence area 2000–2008 (RLPB records).**

| Year | No. of Flocks | Total Doses | Max. Doses/ Flock | Avg. Doses/ Flock | Flocks >2000 Doses/yr | % Flocks >2000 Doses/yr |
|------|------|------|------|------|------|------|
| 2000 | 55 | 83,700 | 7,700 | 1,522 | 14 | 9.9% |
| 2001 | 184 | 385,400 | 19,700 | 2,095 | 53 | 18.2% |
| 2002 | 431 | 519,450 | 13,600 | 1,205 | 78 | 11.9% |
| 2003 | 611 | 472,550 | 8,000 | 773 | 49 | 5.7% |
| 2004 | 622 | 374,600 | 6,400 | 602 | 30 | 3.6% |
| 2005 | 536 | 337,800 | 8,000 | 630 | 26 | 3.6% |
| 2006 | 524 | 311,350 | 8,400 | 594 | 27 | 4.0% |
| 2007 | 319 | 179,700 | 4,750 | 563 | 16 | 4.4% |
| 2008 | 46 | 19,100 | 1,500 | 415 | 0 | 0.0% |
| Total 2000–08 | 1,320 | 2,683,650 | 37,900 [a] | 2,033 | 66 [b] | 5.0% [b] |

**Table 10. Number of flocks vaccinated in the high prevalence area 2007–2012 (Pfizer database).**

| Year | No. of Flocks | Total Doses | Max. Doses/Flock | Avg. Doses/ Flock | Flocks >2000 Doses/yr. | % Flocks >2000 Doses/yr |
|---|---|---|---|---|---|---|
| 2007 | 755 | 543,700 | 7,350 | 720 | 59 | 7.8% |
| 2008 | 1,328 | 980,250 | 10,000 | 738 | 104 | 7.8% |
| 2009 | 1,321 | 950,450 | 12,000 | 719 | 91 | 6.9% |
| 2010 | 1,316 | 897,350 | 8,500 | 682 | 80 | 6.1% |
| 2011 | 1,384 | 1,027,350 | 29,300 | 742 | 101 | 7.3% |
| 2012 | 1,288 | 1,039,600 | 11,000 | 807 | 106 | 8.2% |
| Total 2007–12 | 2,483 | 5,438,700 | 33,000 [a] | 2,190 | 116 [b] | 4.7% [b] |

Tables 9 & 10. The number of flocks vaccinated and number of doses/flock/year in the NSW HPA from 2000–2012. The data in Table 9 were extracted from the RLPB records and the data in Table 10 were extracted from the Pfizer database.

[a] Maximum number of doses/flock 2000–2008 (Table 9) or 2007–2012 (Table 10).

[b] Number and % of flocks where >8000 sheep were vaccinated in the period 2000–2008 (Table 9) or 2007–2012 (Table 10).

annually). The average number of sheep vaccinated per flock annually rose from 1522 in 2000 to 2095 in 2001 before declining to 1205 in 2002 and levelling off at ~600-800/year from 2003–2012. The percentage of these flocks with more than 2000 sheep vaccinated annually ranged from 10–18% from 2000 to 2002, reduced to ~4–6% of from 2003–2006 then rose to 6–8% from 2007–2012 after sales were permitted through rural merchandisers.

## Sheep breeds affected

**On-farm investigations 1982–2006.** Analysis of 779 OJD positive laboratory submission from field investigations confirmed 91% of the 642 submissions with breed category defined were Merino flocks.

**Abattoir monitoring NVDs 2003–2007.** A subset of 2,249 National Vendor Declarations associated with abattoir consignments from 2003 to 2007 (sheep ≥2 years of age) were identified where the sheep breed was recorded as either Merino or Crossbred/British Breed.

Merinos comprised 89.3% of consignments, 93.8% of 616,500 sheep inspected, 94.7% of 1036 positive consignments, 96.1% of 340,940 sheep in positive consignments and 97.5% of 6904 sheep with lesions attributable to OJD. Merinos also showed a higher proportion of heavily infected consignments with 14.6% (143/981) of consignments detected with ≥5% lesions, 4.5% (44/981) with ≥10% lesions and 0.2% (2/981) exceeding 30% lesions. This compared with positive consignments involving Crossbred/British Breed sheep where 3.6% (2/55) were detected with ≥5% lesions while none exceeded 10% lesions.

## Discussion (See also S7 Discussion in S1 File)

Controlling Johne's disease is economically important for the animal production industries and for the veterinary community. In the event that MAP is established to be zoonotic, controlling Johne's disease will become important for society as a whole. Despite pilot studies showing Johne's disease can be controlled with herd, animal and feces management [29], the increasing prevalence of Johne's disease in cattle [2] and sheep [44] suggests these controls are currently ineffective in practice and this remains a serious cause for concern. Spread of infection within and between flocks and herds appears to continue despite voluntary control programs whose aims are to restrict exposure of susceptible animals to infected animals and feces.

There are no economically feasible or effective antibiotics, or similar therapeutic agents, currently available for treating MAP infection in sheep. Notwithstanding, vaccination is a potentially valuable adjunct to the control of OJD because fecal shedding of MAP by infected

animals is reduced, thereby reducing environmental contamination and exposure of uninfected animals. This study provides evidence that vaccination can be used to control OJD over wide geographic regions.

### Importance for the Merino wool and sheepmeat industry (See also S7A Importance for the Merino wool and sheepmeat industry in S1 File)

This manuscript extends prior observational reports [44, 51] on monitoring for OJD in Australia. Abattoir monitoring has been demonstrably cost efficient and is particularly important for the Merino wool and sheepmeat industry in Australia [44]. In 2009, it was estimated that there were 77 million sheep in Australia with 26.1 million (34%) in NSW of which 21.9 million (84%) were merinos. There are numerous anecdotal, but few published reports, citing merinos in general, and superfine wool merinos in particular, as being more susceptible to OJD than first-cross or British-bred sheep [45, 85–87]. This hypothesis is supported by the analysis of the breed distribution in the present study (see S7A Importance for the Merino wool and sheepmeat industry in S1 File for more detail).

### Economic and social impact of OJD (See also S7B Economic and social impact of OJD in S1 File)

The economic and social impact of OJD on the sheep industry in Australia from 1981 to 2009 was substantial [88, 89]. Prior to the availability of vaccine in 2000 there was no effective means of limiting the spread between and within flocks. Regulation (quarantine) placed a heavy burden on known or suspected infected properties. In addition, movement restrictions between prevalence areas led to severe impacts on trading opportunities for all flocks in affected regions.

At the same time, many infected flocks had a high annual mortality rate ($\geq$5%) [58, 88–91] evidenced by the demand for vaccination between January 2000 and April 2002 prior to registration of the vaccine for widespread use [58].

### Factors in disease spread (See also S7C Factors in disease spread in S1 File)

Abattoir monitoring was critical in identifying sub-clinical disease in flocks where the owner had no other evidence of disease. This overcame the major impediment to halting unwitting spread of infection, both locally and to new regions, critical to an effective disease control program. The producer workshops and PDMPs contributed significantly to producer understanding of the epidemiology of OJD.

### Cost of vaccination (See also S7D Cost of vaccination in S1 File)

The cost of vaccination was also a significant imposition on producers at a time when they were receiving low prices for their sheep. Combined with the effects of drought, an aging farming population, the increasing cost of farm inputs and low prices for sheep and wool, the added burden of OJD encouraged many producers to leave the sheep industry.

### Abattoir monitoring–disease surveillance and vaccine driver

Abattoir surveillance is both sensitive and specific [58, 75–77, 83] and in NSW it has provided detailed and accurate disease surveillance data, critical for effective disease control over any large geographical region. At the same time, the abattoir surveillance data from this Government and industry partnership program has been crucial to this study of the effectiveness of vaccination for OJD.

Prior to commencement of abattoir monitoring in late 1999 and the initiation of vaccination in 2000, particularly in the HPA, the flock and animal-level prevalence of OJD in NSW had been increasing exponentially [42–44, 58, 92]. The first two months of abattoir monitoring (Nov.-Dec.1999) showed an initial peak in percentage of positive sheep from all consignments from the HPA (3.9%). This coincided with a period (late Spring, early Summer) of normally high annual turnoff of cull sheep. It was also a time when many flocks were suffering increasing clinical losses and were culling heavily infected mobs. Subsequently, animal-level prevalence in the HPA fluctuated from 2.4% in 2000 to 1.8% in 2001 before increasing to an annual peak of 2.6% in 2002. It then progressively declined to 0.8% by 2009 (Fig 7B).

This increase after the first two years of vaccination is not unexpected as access to vaccine was restricted to heavily infected flocks and only a portion of the population (primarily the newborn lambs) were vaccinated each year. In the meantime, OJD was at continued risk of spreading in the remaining unvaccinated population. Additionally, the long incubation period of OJD before detectible thickening of the terminal ileum (often 2 years or greater) dictates that sheep be at least 2-years-old to warrant monitoring at slaughter, ie young adult sheep not lambs. Breeding ewes or wool producing sheep (wethers) would normally be retained for 4–6 years to maximize productivity, while lambs/hoggets for meat production would normally be sold when they are 12-18-months-old. Hence, it would be a minimum of two years before those vaccinated as lambs were included in the animals monitored at slaughter, with increasing numbers by 4–6 years after vaccination.

Concurrently, the impact of drought in NSW generally, and the HPA in particular, from 2002 onwards led to accelerated culling of non-breeding stock (wethers for wool production) and older age groups of ewes (frequently unvaccinated). The selective culling of more heavily OJD infected mobs showing clinical disease (wasting or mortalities) may have contributed to a surge in detection from 1999–2004, particularly consignments with 10% or more lesions attributable to OJD (Fig 9A & 9B). At the same time, culling high risk mobs would have reduced (at least temporarily) the average animal-level prevalence within these flocks.

The initial shortage of vaccine and restricted usage pending final registration in April 2002 [58] would also have delayed the overall impact of vaccination in the HPA. Financial stringency due to the drought, balanced by financial assistance to purchase vaccine, was also a significant factor in the uptake of vaccination.

Thereafter, however, there was a progressive area-wide decrease in animal level disease prevalence in the HPA, as would be expected in an effective vaccination campaign. This was despite the flock-level prevalence continuing to remain very high, with more than 50% of HPA flocks estimated to be infected in 2008 based on abattoir monitoring (median 51.8%; 95% upper confidence limit– 95.6%) [51].

Concurrently with the marked decline in animal-level prevalence of sub-clinical disease evidenced by abattoir monitoring, RLPB District Veterinarians by 2007 were reporting that the level of clinical disease and mortality rates attributable to OJD within vaccinating infected flocks had also decreased markedly [60]. This decrease in clinical disease on-farm was consistent with the 90% reduction in mortality reported in the intensive vaccine trials [34].

## Vaccinating infected sheep–concerns regarding exacerbation of disease (See also S7E Vaccinating infected sheep–concerns regarding exacerbation of disease in S1 File)

The regulatory authorities had expressed concern that vaccinating infected animals would exacerbate the progression of disease. The preliminary findings in the "Whole-of-Flock" vaccination trial [93] in late-2000 were crucial to enable the initial recommended vaccination age

for lambs in infected flocks to be safely extended from 4–12 weeks to 16 weeks to better fit in with routine lamb marking procedures.

At the same time it enabled approval of policies to safely permit vaccination of older sheep in known infected flocks, including adult mobs known to be heavily infected. **It is important to recognize how crucial this trial was in enabling a vaccination program unrestricted by age group or history of exposure [93].**

## Vaccine–preventative or therapeutic? (See also S7F Vaccine–preventative or therapeutic? in S1 File)

It remains unclear whether vaccination for OJD has a therapeutic effect (ie delay in progression of disease in infected animals or regression of pre-existing pathological changes), or whether the primary benefit is a high level of protection following vaccination prior to exposure or establishment of infection [94]. It is likely in heavily infected flocks that a high proportion of lambs are exposed to infection before they are vaccinated.

The results from the "Whole of Flock Vaccination Trial" [93] would support the concept of a therapeutic effect, as the great majority of vaccinates were either sub-clinically infected or exposed to MAP prior to being vaccinated, even those vaccinated as lambs.

Alonso-Hearn et al (2012) [30] suggested a therapeutic effect for a heat killed "whole cell" vaccine in dairy cattle in Spain (Silirum® - Zoetis, USA; CZ Veterinaria, Porriño, Pontevedra, Spain). Silirum®, like Gudair®, contains 2.5mg/ml of bovine MAP strain 316F as the sole active ingredient. They reported a significant attenuation of pre-existing infection in cows naturally infected with paratuberculosis that were adults at the time of vaccination [30].

## Age groups vaccinated

The primary message was to vaccinate as lambs (75% of vaccine usage 2000–2002 and 94% from 2007–2009) and preferably <4 months of age ("approved vaccinates"– 41% from 2000–2002).

However, many producers throughout the period of this study, particularly those suffering heavy losses, were found to have vaccinated their whole flock or selected older age groups (17% hoggets and 8% adults from 2000–2002 falling to 2% and 4% of vaccine sales in 2007–2009). Approval to vaccinate older sheep had been restricted judiciously prior to registration of the vaccine in April 2002 due to the limited number of vaccine doses available from the manufacturer.

The decline in vaccination of older sheep is likely attributable to the progressive rise in fully vaccinated flocks, the decline in the number of heavily infected consignments/flocks identified, and the early detection of new infection through abattoir monitoring.

## "Herd" immunity (See also S7G "Herd" immunity in S1 File)

Vaccination of older lambs, hoggets and adult sheep undoubtedly contributed to a more rapid increase in "Flock/Herd Immunity", both within individual flocks and the general sheep population of the HPA. This outcome is evidenced by the "cumulative" vaccination status in the HPA reaching an estimated 16% by 2004, 45% by 2006 and 70% in 2009 (Table 8) [84], the target commonly espoused for an effective disease control program based on vaccination.

This result is consistent with a previous estimate that ~70% of replacement sheep were being vaccinated annually in the HPA by 2007 [60]. Vaccination of older sheep is highly likely to have contributed to a more rapid decline in animal-level prevalence of OJD than might otherwise have been observed [94, 95].

## Vaccinating flocks in the HPA

The high initial annual vaccination rate per flock (av. 1,475 for 2000–2002) was attributable to the restriction of vaccine to heavily infected flocks, and the availability of financial assistance for eligible flocks. These factors provided increased incentive to vaccinate older age groups in addition to lambs. The persistent drought, and reduction in average flock size, may also have contributed to subsequent declining numbers vaccinated annually per flock (~600-800/year from 2003–2012, Tables 9 and 10).

## Persistent flock infection/excretion

Persistent flock infection was confirmed in the present study where animal-level prevalence in the HPA declined from 2.4% in 2000 to <1% from 2005–2009 (Fig 7B) while the infected flock-level prevalence (based on % of consignments positive) remained above 33% from 2001 to 2009 (Fig 7A).

These results are consistent with the findings of Windsor (2014) who reported that 81% (30/37) of known infected commercial flocks monitored in a research trial using PFC in NSW and Victoria continued to shed MAP. This was despite significant declines in estimated OJD prevalence following vaccination as lambs for ≥5 years [96]. They also reported a decline in animal-level prevalence from a pre-vaccination median of 2.7% (95% confidence interval: 1.4–6.9) to a post-vaccination median of 0.7% (0.4–1.3), a similar decrease to that reported for the HPA in the current study. They considered these flocks at risk of spreading disease or suffering recrudescence of losses if vaccination were to cease and they advised flock managers to persist with vaccination [5, 56, 96].

## Factors contributing to a decrease in regional animal-level prevalence

The decrease in OJD regional animal-level prevalence in the HPA that we observed from 2003 onwards could have resulted from causes other than vaccination. For example, NSW was subject to an unusually long and severe drought during the study period. Termed the "Millennium" drought it ran from 2002 to 2009. However, there is some evidence to refute this alternative interpretation of the data.

In contrast to the fall in OJD animal-level prevalence (and static flock-level prevalence) in the NSW HPA, there was a marked and progressive increase in the flock-level prevalence of OJD in the adjacent state of Victoria during the period leading up to 2008–2009 [51]. The drought was equally severe in Victoria as in NSW during this time. Conversely, vaccine sales in Victoria were low and abattoir monitoring was conducted on a more limited basis than in NSW [51].

We conclude that the decrease in OJD that we documented in NSW cannot solely be ascribed to diminished rainfall. In fact hand feeding of hay and grain from the ground due to the drought could have increased opportunities for infection by the feco-oral route. Further, the decrease in OJD in the HPA of NSW was not replicated in the MPA where vaccination was progressing more slowly.

Improved management practices aimed at OJD control could also explain, in whole or in part, the observed fall in animal level prevalence in the HPA. The provision of financial assistance in 2002–03 was contingent on development of a formal PDMP tailored for each property. These were aimed at promoting management practices associated with improved on-farm control of OJD, in addition to the widespread application of funding to the purchase of vaccine. It is likely that vaccination and improved management yielded a complementary benefit [93]. Certainly, an improved understanding of age susceptibility was incorporated into management practices to minimize exposure of highly susceptible lambs to pasture contaminated by adult sheep more likely to be excreting MAP.

It is unlikely that many producers implemented targeted on-farm management and control strategies prior to OJD being confirmed in their flock. From 2000 onwards, abattoir monitoring progressively assumed greater importance in confirming the presence of OJD in individual flocks. It complemented (then replaced) more targeted (but costly) surveillance strategies such as investigation of clinical disease and the testing of flocks identified following tracing of sheep movements or identified as neighbors of known infected flocks.

## Prevalence areas–role of abattoir monitoring

During the period of this study, the areas that were declared as HPA and MPA for OJD were progressively modified [51, 53, 58, 59, 72], with areas of lower risk transitioning to higher risk following increased abattoir detection of OJD-infected consignments. Vaccination of sheep in the MPA was not permitted until 2003 and was uncommon until 2007. The animal-level prevalence of OJD was only observed to decline in areas where vaccination was common–the NSW HPA. The HPA was also the area where the feedback of abattoir OJD surveillance data to sheep producers was the most effective in demonstrating the severity of the problem of sub-clinical disease.

A primary aim of the abattoir monitoring is to advise producers in a timely fashion of the results of their monitoring. This advice is particularly important for those infected flocks where abattoir monitoring can detect infection when there is no evidence of clinical disease. Feedback to producers also provides quantitative information on the level of infection in the consignment monitored. This promotes the uptake of vaccine and PDMPs in newly detected flocks, while also providing a measure of the progress in controlling sub-clinical infection in those flocks already vaccinating.

In the LPA, the primary aim of abattoir monitoring is early detection of sub-clinical infection so that the risk to other flocks can be promptly neutralized [97]. Concurrently, notification of negative monitoring results provides confidence to individual producers that (a) their flocks are being monitored, (b) their results are negative, and (c) disease control measures based on prevalence areas and biosecurity are built on sound principles.

## NLIS and traceability to property of origin

The results of this study also demonstrate many of the benefits available from the NLIS livestock tracing system now in place in Australia. In fact, the need for improved traceability required by abattoir monitoring for OJD [75] accelerated the implementation of NLIS for sheep. It also provided a comprehensive audit mechanism of PIC details recorded on National Vendor Declarations at slaughter [59].

The NLIS system (eartag applied to lambs on the property of birth) enables every animal slaughtered in Australia to be traced back to the consignment property, as well as all other properties on which the animal has been resident. This level of information is essential when conducting tracing of long incubation diseases. Importantly, the NLIS facilitates data feedback to the owner, which alerts them to the disease status of their animals and promotes rational and logical management decisions on how best to control diseases such as OJD.

In particular, NLIS enabled vaccination programs to be targeted at infected flocks and allowed OJD prevalence areas to be accurately defined. This in turn permitted risk-based trading systems to be implemented, without unnecessary restriction of animal movements [44].

## Sensitivity and specificity (See also S7H Sensitivity and specificity in S1 File)

The sensitivity and specificity of abattoir monitoring was unknown at the commencement of monitoring in 1999. However, it was clear within weeks that the procedure was capable of

detecting sub-clinical OJD (often in a high proportion of animals) in many consignments from the HPA, while consignments from the LPA were routinely negative.

A research trial was incorporated into the routine abattoir monitoring in 1999–2000, to determine the sensitivity and specificity of visual/tactile inspection of viscera for lesions suggestive of OJD, with follow-up histopathology on 3 animals [75]. This study estimated consignment sensitivity of 97%, and a practical working sensitivity of 90%, in consignments of >300 adult sheep, for any sheep population in which OJD had been recognized for many years (see S7H Sensitivity and specificity in S1 File for details).

Nonetheless, some reports highlighted perceived limitations of abattoir monitoring. These included concerns about low sensitivity of visual inspection [55], concerns which were subsequently allayed by a second research trial conducted in April 2002 in one of the NSW abattoirs (Goulburn) participating in the current study [76, 77]. This confirmed individual animal sensitivity of abattoir inspection compared with histopathology of 74–87% (depending on the inspector), and specificity of 97–98% in high prevalence consignments. Individual animal sensitivity was estimated at ~50% in low prevalence consignments (including infected consignments from low prevalence areas).

Reports from both trials emphasized the importance of maximizing sensitivity and specificity by using experienced qualified meat inspectors who had undergone specific training in inspection, sampling and tracing procedures for OJD in the abattoir [75–77]. All inspectors participating in the current study satisfied these requirements.

A review of histopathological findings from NSW and interstate abattoirs in the present study confirmed the high specificity of visual inspection for detection of OJD in the HPA (82%). In contrast, the specificity was 18% in the LPA where inspectors were deliberately encouraged to sample anything remotely suspicious of OJD [83].

There remains no clear scientific explanation as to why the majority of sheep sampled (83%) exhibit multibacillary pathology while others in the same consignment exhibit paucibacillary pathology. Variable individual genetic resistance factors within a flock [5], exposure to differing infective doses/timeframes, or the stage of infection are possible explanations. It is clear that inspectors are capable of detecting both forms of pathology, with no obvious gross or microscopic differences other than the overt presence or absence of acid-fast organisms on histopathological examination [83].

## Limitations

Classifying consignments with paucibacillary pathology as OJD positive where the property had a prior or subsequent (within 2 years) OJD confirmation is justified as a practical way of categorizing the data for a disease with a long incubation period (years) and where a relatively high percentage of cases are paucibacillary. Using these criteria there were only 178 (0.6%) consignments from NSW culled from the dataset due to unresolved status, and only 17 (0.1%) of these consignments derived from the LPA where a positive detection is of more critical importance.

Limiting the number of sheep to be sampled to a maximum of three per consignment was dictated by economic considerations, and was supported by the high individual animal sensitivity as evidenced in the abattoir sensitivity trials [75–77]. While increasing the sample size to 6 or 9 [55] would have increased flock-level sensitivity in low prevalence flocks and the LPA in particular, it would have markedly increased the cost. This issue was dealt with for consignments from the LPA by encouraging sampling of any viscera showing signs remotely suggestive of OJD while retaining a maximum sample size of three.

Similarly, concerns that other conditions grossly resembling OJD would reduce sensitivity by displacing samples with OJD [55] was found not to be a major issue. Even at 10 sheep per

minute, inspectors proved highly skilled in differentiating subtle changes due to early cases of OJD from normal intestines and changes due to other conditions. This was a direct result of their training and the inspection of large numbers of viscera on a daily basis.

Inspectors would put up to three viscera aside for later sampling, but often replaced them with more advanced examples as inspection progressed. Most difficulty was reported with inspection of lambs and hoggets under 2 years of age, where the intestinal immunity was actively maturing (evidenced by slight thickening of the terminal ileum) in response to internal parasites and common intestinal flora. In light of the long incubation period of OJD, this issue was minimized by restricting OJD inspection to sheep ≥2 years of age to maximize sensitivity, specificity and cost effectiveness. Abbott and Whittington (2003) discuss in detail factors likely to influence sensitivity and specificity [55].

Reducing the tissues processed from three initially (terminal ileum, ileo-cecal valve and adjacent mesenteric lymph nodes) to a single tissue per animal (terminal ileum) in 2002 enabled samples from up to 3 animals to be processed per microscope slide. This reduced the laboratory costs by up to two-thirds with minimal loss of sensitivity (unpublished report to Animal Health Committee Sub-Committee on Animal Health Laboratory Standards—Sept. 2002).

The number of infected animals per consignment was estimated by pro-rata "correction" of the number of sheep detected with lesions, according to the proportion of samples positive for OJD on histopathology (For more details refer to S5 Inspection, sampling & data collection in S1 File).

This was premised on a random distribution of OJD positive animals amongst those detected with gross lesions in the consignment, and that the animals sampled were representative of the animals with lesions. This was considered the most reasonable, practical and economical approach to standardizing the data and was applied consistently throughout the study.

## Support by abattoir management

The voluntary cooperation of abattoir management was crucial in facilitating the activities of the inspectors, including provision of safe sample handling facilities and supply of NLIS and PIC/Owner information. They also played a critical role in minimizing the risk, during transport and lairage, of sheep from another consignment being accidentally mixed with the consignment to be monitored.

This support was critical, as there was zero tolerance for error in property and sample identification, as any mistaken identity would have wrongly penalized the producer and rapidly discredited the program. Abattoir management, in turn, stood to benefit by an overall improvement in sheep health and the improved traceability (provided by the NLIS) being progressively demanded by export and domestic markets.

## Producer acceptance (trust) of abattoir monitoring results

Initially, abattoir monitoring was used as a screening test with follow-up on-farm confirmatory testing (serology and histopathology). However, as producer confidence in the results increased and the impact of a positive diagnosis decreased (vaccination, financial assistance, improved trading opportunities), the majority of producers in the HPA (and the MPA) accepted the abattoir monitoring result as definitive, further reducing the cost of on-farm confirmatory testing.

Positive (or inconclusive) abattoir monitoring results from the LPA were always followed up by on-farm risk assessment and appropriate testing to confirm or allay suspicion [97].

### Saleyard consignments (See also S7I Saleyard consignments in S1 File)

Concerns were raised about producers avoiding monitoring by selling through saleyards and their sheep being slaughtered in mixed consignments from multiple vendors [43, 52]. This was considered a factor in missed detection of some smaller individual infected flocks, however the impact was minimized with the cooperation of the saleyard and transport operators. The integrity of the majority of larger individual property consignments sold through saleyards was able to be carefully maintained up to the time of slaughter. This segregation enabled them to be classified as "direct" single property consignments and included in the study. This approach was particularly important for regions distant from an abattoir where it was more cost effective for producers to sell through saleyards.

### Regional prevalence data and surveillance

The ability to monitor very large numbers of direct consignments from all sheep producing regions of NSW throughout the year provided comprehensive detailed regional prevalence data. Identification of the precise location of positive flocks was critical for the review and placement of prevalence area boundaries [65, 92]. The aim was to maximize control over infected flocks while minimizing the regulatory impact on trading for flocks considered unlikely to be infected.

Ongoing and effective disease surveillance is necessary to be able to determine the efficacy, or lack thereof, of control interventions such as vaccination. More importantly, ongoing and effective disease surveillance is necessary to be able to rapidly provide sheep producers with evidence of recent infection in their flocks. It is likely that the lag phase in detection, due to the long incubation period of OJD, has most impact in regions where OJD is newly emerging.

However, the comprehensive centralized monitoring afforded by abattoir surveillance, offers the best opportunity for early and more cost effective detection compared with more targeted, labor intensive, on-farm surveillance procedures. This evidence also provides the best motivation for a producer to initiate vaccination–well before OJD mortality rises to a level where there is high level and widespread environmental contamination on the property due to increased infection rates.

### Accidental operator self-inoculation

Minimizing the risk of accidental self-inoculation with Gudair® was also a primary aim of the vaccination program. The risk was mitigated through a concerted effort by Pfizer, NSW DPI, RLPBs and the sheep industry to communicate the nature of the risk through training programs for producers and vaccine merchandisers, and through standardized vaccine handling and vaccination procedures.

The development by Pfizer of the novel "Sekurus"® vaccinator syringe with a protective shroud and double locking mechanism, and increasing the awareness within the rural medical community were critical initiatives. Many producers minimized their personal risk by employing contract vaccinators.

Nonetheless small numbers of producers suffered severe impact from accidental self-inoculation [98, 99] and fear of self-inoculation caused some producers to avoid vaccinating their flocks.

### Vaccine abscesses (See also S7J Vaccine abscesses in S1 File)

The risk of vaccine abscesses [34] was also of serious concern to the sheep industry both from an animal welfare perspective and the potential for abscesses to impact on abattoir processing.

Producer workshops and a standardized vaccination procedure requiring sub-cutaneous injection high on the neck (at the base of the ear) were effective in minimizing untoward outcomes. From commencement of monitoring in 1999, producers were notified directly by the abattoir if any abscesses were detected in neck muscle at slaughter.

## Potential role of vaccination in controlling BJD in cattle (See also S7K Potential role of vaccination in controlling BJD in cattle in S1 File)

The endemic spread of BJD internationally in recent decades has been recently likened by Whittington *et al.* (2019) to a slow moving, uncontrolled long-incubation pandemic [4]. Vaccination with a killed vaccine may be a valuable adjunct to current test-and-cull strategies for the control of Johne's disease in cattle. Government agencies, however, have been reluctant to initiate Johne's disease vaccination programs in cattle. In the past, the most compelling concern was loss of ability to diagnose tuberculosis (usually *M. bovis*) using skin testing [4, 100]. Out of 22 countries with formal BJD control programs, only 7 were reported to have included the option of vaccination [4]. This is despite the evidence that standard pasteurization reliably kills *M. bovis* in bovine milk, in contrast to MAP which is not reliably killed by pasteurization [101, 102].

In Australia, where bovine tuberculosis has been eradicated, vaccination of infected dairy or beef herds against paratuberculosis is considered likely to be cost-effective [4].

We conclude that Government regulatory agencies and the veterinary profession should give serious consideration to progressing further Johne's disease vaccination studies in cattle and to comparing vaccination cost and efficacy with failure to vaccinate [4, 30, 32, 33, 38, 103]. An effective vaccine could make a major contribution to control of the production losses and potential human health risks associated with BJD.

## Potential role of abattoir monitoring for BJD in cattle

Similarly, the potential surveillance benefits of implementing abattoir monitoring for BJD in cattle (particularly dairy cattle) should be explored. In a 2019 review, 38 countries out of 48 were reported to have formal tuberculosis control programs [4]. While cattle at slaughter throughout the world are routinely monitored by inspectors for tuberculosis, it is extremely rare for concurrent monitoring to be undertaken for MAP infection [104].

Dahmen et al (2018) recently reported visual detection of 143 cases of BJD during official meat inspection between 2015 and 2017 in an abattoir in Germany [105]. They identified both early and advanced cases of MAP infection (confirmed by histopathology). They also demonstrated the presence of MAP by culture and PCR in skeletal muscle from 39% of the 143 BJD positive cattle.

Deer in New Zealand are routinely monitored at slaughter for both tuberculosis and paratuberculosis (primarily due to C-strain) in a producer funded program [4, 106].

If MAP is zoonotic, cattle (through milk, milk products and meat) likely present a greater risk than sheep for food-borne infection of humans around the world [5, 20], although milk or meat from sheep, goats and deer may present a risk in many countries. In contrast to tuberculosis, only 22 of 48 countries were identified with formal BJD/paratuberculosis control programs, with major surveillance deficits impacting their ability to implement effective disease control [4]. Routine monitoring of cattle (deer and goats) for paratuberculosis at slaughter, in conjunction with current official monitoring for tuberculosis, would be a cost-effective approach to resolving the surveillance deficit for BJD. As demonstrated in the current study, it would facilitate producer understanding of the extent of infection in their herds and promote more effective control measures including vaccination.

### Animal model for mycobacterial diseases in humans

Vaccination of sheep against MAP with Gudair®, a killed whole cell vaccine with oil adjuvant, may also provide an important animal model for vaccination for mycobacterial diseases (tuberculosis, leprosy and possibly Crohn's disease) in humans. While noting the adverse outcome of severe granulomatous reactions typically seen in humans following accidental self-inoculation with Gudair®, alternative adjuvant and delivery mechanisms should be explored.

## Conclusions

Results of the detailed analysis of more than 12 million vaccinations and 7 million carcass examinations presented in this study are consistent with the hypothesis that vaccination against OJD contributed to reversing the previously inexorable increase in animal-level prevalence of OJD in NSW, Australia where more than 35 million sheep were at risk of infection in 2000. The data support the wider use of vaccination to combat OJD.

We further conclude that these data support intensive evaluation of the use of vaccination against MAP in other species in which MAP causes or may cause disease.

Finally, we propose that disease control to $\leq$ 1% OJD animal-level prevalence can be achieved by the combination of voluntary vaccination, abattoir monitoring, feedback of surveillance data to sheep producers and the implementation of appropriate disease management strategies on farm.

## Supporting information

**S1 File.**
(DOC)

## Acknowledgments

The program in NSW was coordinated and managed under the auspices of NSW DPI. The National OJD Program was capably managed and coordinated through Animal Health Australia (Lorna Citer, Manager Endemic Diseases) and delivered through NSW DPI (veterinarians, livestock officers and the many committed staff members). The project could not have proceeded without the commitment and diligence of the OJD abattoir inspectors (including Tony Ware, Jim Smith, Tony Hogben, Wayne Gilbert, Gary Murphy. Mark Lyons, John McKinnon, Stuart Bray, Ian Brown and many others). The support of management at export abattoirs (Dubbo, Goulburn, Deniliquin & Wallangarra), domestic abattoirs (Cowra, Cootamundra, Young, West Wyalong, Junee, Mudgee and Griffith) and interstate abattoirs (Ararat, Brooklyn, Cranbourne, Lobethal, Murray Bridge and Port Pirie), together with the Australian Quarantine Inspection Service (AQIS), was essential.

DVs and Animal Health Officers with RLPBs provided ongoing support notifying producers, undertaking follow-up field investigations and promoting the uptake of vaccination. The contributions from the data management team at Wagga Wagga (Tracey Kingham, Jenny Fury, Raylene Heath & James Hamilton), and Orange (Maurie Ryan), and pathologists at the Regional Veterinary Laboratories at Orange and Menangle, are gratefully acknowledged.

We would like to thank the sheep industry representatives who contributed to the development of state and national disease control policies, often under very difficult circumstances, and the many producers who embraced vaccination and implementation of PDMPs for improved control of OJD in their flocks.

The Australian sheep industry owes a great debt of thanks to Dr. Esteban Rodriguez Sanchez (Technical Director) and Dr. Eugenia Puentes (I&D Director) of CZ Veterinaria SA, who

developed and evaluated a killed oil-adjuvant vaccine (based on bovine MAP strain 316F) for the control of clinical Johne's disease in sheep and goats in Spain in the early 1990s. This vaccine was subsequently commercialized as Gudair® by CZ Veterinaria SA, who responded with great commitment from 2000 onwards to the unprecedented demand for OJD vaccine from Australian sheep producers.

The assistance of Dr. Arieh Bomzon in the final editing of the manuscript is greatly appreciated.

We also acknowledge the on-going behind the scenes support provided by our families during this extended disease control program and the subsequent preparation of this report.

Co-author Marilyn Evers sadly passed away on 28 July 2019. She was a driving force in liaising with producers and developing coordinated, practical policies for the control of OJD in NSW and across Australia.

## Author Contributions

**Conceptualization:** Ian J. Links, Laurence J. Denholm, Marilyn Evers, Lloyd J. Kingham.

**Data curation:** Ian J. Links, Laurence J. Denholm.

**Formal analysis:** Ian J. Links, Robert J. Greenstein.

**Funding acquisition:** Ian J. Links, Laurence J. Denholm, Marilyn Evers.

**Investigation:** Ian J. Links, Laurence J. Denholm, Lloyd J. Kingham.

**Methodology:** Ian J. Links, Laurence J. Denholm, Marilyn Evers, Lloyd J. Kingham.

**Project administration:** Ian J. Links, Laurence J. Denholm, Marilyn Evers, Lloyd J. Kingham.

**Resources:** Ian J. Links, Laurence J. Denholm, Marilyn Evers, Lloyd J. Kingham.

**Software:** Ian J. Links.

**Supervision:** Ian J. Links, Laurence J. Denholm, Marilyn Evers, Lloyd J. Kingham.

**Validation:** Ian J. Links, Laurence J. Denholm, Robert J. Greenstein.

**Visualization:** Ian J. Links, Marilyn Evers, Robert J. Greenstein.

**Writing – original draft:** Ian J. Links, Robert J. Greenstein.

**Writing – review & editing:** Ian J. Links, Laurence J. Denholm, Lloyd J. Kingham, Robert J. Greenstein.

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
