## [Decision Letter · Decision Letter 0]

8 Feb 2021

PONE-D-21-01240

Is Vaccination a Viable Method to Control Johne’s Disease Caused by Mycobacterium avium subsp. paratuberculosis? Data from 12 Million Ovine Vaccinations and 7.6 Million Carcass Examinations in New South Wales, Australia from 1999-2009

PLOS ONE

Dear Dr. Links,

Thank you for submitting your manuscript to PLOS ONE. After careful consideration, we feel that it has merit but does not fully meet PLOS ONE’s publication criteria as it currently stands. Therefore, we invite you to submit a revised version of the manuscript that addresses the points raised during the review process.

Your manuscript has been reviewed by two experts in your field. A revision is necessary before  a decision can be made. Please follow the reviewer comment  to make the necessary revision or give a rebuttal.

We look forward to receiving your revised manuscript.

Kind regards,

Yung-Fu Chang

Academic Editor

PLOS ONE

2. In your Methods section, please provide the name of the slaughterhouses where the animals were sacrificed.

Reviewers' comments:

Reviewer's Responses to Questions

**Comments to the Author**

1. Is the manuscript technically sound, and do the data support the conclusions?

Reviewer #1: Yes

Reviewer #2: Yes

2. Has the statistical analysis been performed appropriately and rigorously? 

Reviewer #1: Yes

Reviewer #2: Yes

3. Have the authors made all data underlying the findings in their manuscript fully available?

Reviewer #1: Yes

Reviewer #2: Yes

4. Is the manuscript presented in an intelligible fashion and written in standard English?

Reviewer #1: Yes

Reviewer #2: Yes

5. Review Comments to the Author

Reviewer #1: Paratuberculosis is a major disease problem the world over affecting ruminant livestock. Efforts to control paratuberculosis have been impeded a lack of understanding of the immune response to the different lineages of the pathogen M. a. Paratuberculosis, the inability to detect infected animals, especially at the early stages of infection and the lack of a vaccine that elicits sterile immunity. The authors have prepared a comprehensive review, based on observational information, describing the evolution of a program to control an outbreak of ovine paratuberculosis in sheep in Australia that started in the in the late 1970s. The events in Australia paralleled the emergence of paratuberculosis becoming a major disease problem in other countries. It is difficult to summarize and highlight the events that led to the development of a successful voluntary program to control paratuberculosis, based on the use of a killed vaccine. However, they have succeeded in developing an informative review highlighting the events that led to a successful outcome. A study in Spain pointed to the value of using a killed vaccine for its therapeutic and prophylactic potential. Initial studies demonstrated vaccination of whole dairy herds endemically infected with a killed vaccine reduced the incidence animals progressing to the clinical stage of disease. Success of trial studies showing the efficacy of vaccination led to gradual acceptance of a killed vaccine strategy to bring paratuberculosis under control. Lessons learned through implementation of the vaccination program, as described, are a major contribution of the review. Paratuberculosis remains a threat because the killed vaccine controls but does not clear infection. Demonstration of success provides points to the potential of using a killed vaccine as a first step in controlling paratuberculosis in livestock at the international level.

Reviewer #2: 1. Although there is no restriction on word count, the manuscript should be presented in a reader-friendly way by which is not containing too much detail. For example, in Methods, the vaccination procedures, financial assistance, producer workshops, feedback to producers, and sheep breeds affected can be briefly stated in manuscript then attach the full information as supplementary data if needed. Word deduction is also needed in Discussion section. I understand that authors want to discuss every situation and factor which may lead to the results, but the discussion pattern should be tightly connected to the significance findings in this study with more concise descriptions.

2. In this study, most of the analysis were calculated based on the number of sheep, so the authors might consider moving table 1 and table 3 to supplementary data.

3. Is MAP 316F the dominant strain in Australia sheep suffering from OJD? Is there any cross protection between the vaccine and other MAP strains that might also exist in Australia ruminant flocks?

4. The authors may use statistical analysis to elucidate the correlation among OJD, vaccine doses, and herd immunity.

6. PLOS authors have the option to publish the peer review history of their article (what does this mean?). If published, this will include your full peer review and any attached files.

Reviewer #1: **Yes: **William C. Davis

Reviewer #2: No

---

## [Author Response · Author response to Decision Letter 0]

27 Apr 2021

In preparing this revised version of our manuscript, we would like to thank the two reviewers for their thorough review and helpful comments in order to improve the overall quality of the report. 

Below are our responses to the editor’s requests and the comments of the two reviewers. 

Editor 

Comment 1: The editor requested that the manuscript meets PLOS ONE's style requirements, including those for file naming.

Response: The manuscript has been revised to meet the journal’s style requirements.

Comment 2: The editor requested that the authors included the names of the slaughterhouses where the sheep were humanely killed.

Response: The names of the abattoirs have been included in the revised manuscript (see subsection “Abattoir monitoring” page 13, lines 238-242).

Comment 3: The editor requested that the figure meet PLoS requirements using the PACE digital diagnostic tool.

Response: The figure files have been converted to TIF format and have been checked on PACE before upload.

Reviewer 1

Comment 1: The reviewer commented that the authors succeeded in developing an informative review highlighting the events that led to successful control of paratuberculosis in sheep using a killed vaccine.

Response: The authors thank the reviewer for the compliments on the quality of the informative review and the clarity, concision, and coherence of the report.

Reviewer 2

Comment 1: The reviewer commented that some parts of the manuscript’s text were not reader-friendly, contained excessive details, and some information in the Methods and Discussion sections could be incorporated into “Supplementary Text”.

Response: We critically reviewed the manuscript’s text to determine whether we could comply with the reviewer’s request to improve the manuscript’s readability and still convey an informative message with enough details (see comments of reviewer 1). Longer paragraphs have been split to improve readability.

The following sections have been retained: Vaccination Procedures; Vaccination discussion; Property Disease Management Plans (PDMPs) & Financial Assistance; Sheep Breeds.

The following sections have been shortened by transfer to Supplementary Text: Feedback to Producers (S6); Susceptibility of Sheep Breeds to Infection (S7a).

Comment 2: The reviewer suggested the authors might consider moving table 1 and table 3 to supplementary data. 

Response: Tables 1 and 3 have been retained in the manuscript. Consignments were the primary unit monitored with flock-level prevalence a critical parameter of the study.

Comment 3: Is MAP 316F the dominant strain in Australia sheep suffering from OJD? 

Response: No. For all practical purposes there is only one dominant Sheep S1 strain in Australia [1] for which Gudair vaccine containing MAP 316F (a bovine vaccine reference strain) provides excellent protection. The Telford strain, for which there is a whole genome sequence [2], is representative of the endemic S strain in Australia and New Zealand. This background details with references have been added to the Introduction (Lines 63-66, P5)

Comment 4: The authors may use statistical analysis to elucidate the correlation among OJD, vaccine doses, and herd immunity. 

Response: While statistical analysis may be possible it has not been pursued at this stage. In essence, the large data set for animals monitored and vaccinated should provide strong statistical support for the validity of the conclusions reported. We are hopeful that publication of the manuscript will stimulate offers to undertake statistical analysis – we would be happy to co-operate in such an undertaking.

If there are any additional concerns and/or comments about the revised manuscript, please do not hesitate to contact me. 

We look forward to receiving the editorial decision in the very near future.

Yours sincerely,

Ian J. Links

Corresponding Author

---

## [Decision Letter · Decision Letter 1]

12 May 2021

Is Vaccination a Viable Method to Control Johne’s Disease Caused by Mycobacterium avium subsp. paratuberculosis? Data from 12 Million Ovine Vaccinations and 7.6 Million Carcass Examinations in New South Wales, Australia from 1999-2009

PONE-D-21-01240R1

Dear Dr. Links,

We’re pleased to inform you that your manuscript has been judged scientifically suitable for publication and will be formally accepted for publication once it meets all outstanding technical requirements.

Kind regards,

Yung-Fu Chang

Academic Editor

PLOS ONE

Additional Editor Comments (optional):

Reviewers' comments:

Reviewer's Responses to Questions

**Comments to the Author**

1. If the authors have adequately addressed your comments raised in a previous round of review and you feel that this manuscript is now acceptable for publication, you may indicate that here to bypass the “Comments to the Author” section, enter your conflict of interest statement in the “Confidential to Editor” section, and submit your "Accept" recommendation.

Reviewer #1: All comments have been addressed

Reviewer #2: All comments have been addressed

2. Is the manuscript technically sound, and do the data support the conclusions?

Reviewer #1: Yes

Reviewer #2: Yes

3. Has the statistical analysis been performed appropriately and rigorously? 

Reviewer #1: N/A

Reviewer #2: N/A

4. Have the authors made all data underlying the findings in their manuscript fully available?

Reviewer #1: Yes

Reviewer #2: Yes

5. Is the manuscript presented in an intelligible fashion and written in standard English?

Reviewer #1: Yes

Reviewer #2: Yes

6. Review Comments to the Author

Reviewer #1: The authors have responded to suggestions that improved presentation and conclusions of the rather extensive presentation of findings.

Reviewer #2: (No Response)

7. PLOS authors have the option to publish the peer review history of their article (what does this mean?). If published, this will include your full peer review and any attached files.

Reviewer #1: **Yes: **William C. Davis, PhD

Reviewer #2: No

---

## [Editor Report · Acceptance letter]

26 May 2021

PONE-D-21-01240R1 

Is vaccination a viable method to control Johne’s disease caused by *Mycobacterium avium* subsp. *paratuberculosis*?  Data from 12 million ovine vaccinations and 7.6 million carcass examinations in New South Wales, Australia from 1999-2009 

Dear Dr. Links:

I'm pleased to inform you that your manuscript has been deemed suitable for publication in PLOS ONE. Congratulations! Your manuscript is now with our production department. 

Kind regards, 

on behalf of

Dr. Yung-Fu Chang 

Academic Editor

PLOS ONE